# CTA-Flux: Integrating Chinese Cultural Semantics into High-Quality English Text-to-Image Communities

## Abstract

We proposed the Chinese Text Adapter-Flux (CTA-Flux). An adaptation method fits the Chinese text inputs to Flux—a powerful text-to-image (T2I) generative model initially trained on the English corpus. Despite the notable image generation ability conditioned on English text inputs, Flux performs poorly when processing non-English prompts, particularly due to linguistic and cultural biases inherent in predominantly English-centric training datasets. Existing approaches, such as translating non-English prompts into English or fine-tuning models for bilingual mappings, inadequately address culturally-specific semantics, compromising image authenticity and quality. To address this issue, we introduce a novel method to bridge Chinese semantic understanding with compatibility in English-centric T2I model communities. Existing approaches relying on ControlNet-like architectures typically require a massive parameter scale and lack direct control over Chinese semantics. In comparison, CTA-Flux leverages MultiModal Diffusion Transformer (MMDiT) to control the Flux backbone directly, significantly reducing the number of parameters while enhancing the model's understanding of Chinese semantics. This integration significantly improves the generation quality and cultural authenticity without extensive retraining of the entire model, thus maintaining compatibility with existing text-to-image plugins such as LoRA, IP-Adapter, and ControlNet. Empirical evaluations demonstrate that CTA-Flux supports Chinese and English prompts and achieves superior image generation quality, visual realism, and faithful depiction of Chinese semantics.

## 1 Introduction

Recent advancements in text-to-image (T2I) models have significantly enhanced the quality and diversity of generated images Ramesh et al. (2021); Saharia et al. (2022); Yu et al. (2022). The Stable Diffusion (SD) model Rombach et al. (2022); Esser et al. (2024) stands out among various T2I frameworks due to its remarkable capability in producing high-quality and photorealistic images from complex textual descriptions. SD's diffusion-based generative process Sohl-Dickstein et al. (2015); Ho et al. (2020); Song et al. (2020b) gradually transports noisy input into clean images through iterative denoising steps, ensuring high fidelity and detailed visual outputs. Concurrently, the emergence of the Flux framework Labs (2024) and its active community Labs et al. (2025) has further boosted the development of T2I models. Flux's collaborative, open-source nature enables rapid evolution and community-driven improvements, leading to consistently improved image generation performance and better accessibility for diverse downstream applications.

However, despite these achievements, current T2I models, including those in the Flux community Labs et al. (2025), still face limitations. A prominent challenge is that language-related bias originates from their training datasets, which are predominantly English-centric. Consequently, models often fail to perform well when generating images from non-English text prompts, struggling to capture the unique cultural semantics and nuanced symbols embedded in other languages Liu et al. (2025). This bias manifests primarily in two distribution gaps: the **linguistic feature distribution gap** and the **visual feature distribution gap**.

From the linguistic perspective, significant distribution gaps exist due to inherent language ambiguities and cultural differences. For instance, the English word "crane" can denote both a type of bird and a piece of heavy construction machinery, whereas the Chinese equivalent does not exhibit such polysemy. This results in semantic ambiguity when translating from languages that use words to express these concepts. Additionally, grammatical and structural disparities between languages further exacerbate the semantic mismatches and hinder the multilingual alignment Yang et al. (2025).

Beyond linguistic factors, the visual feature distribution gap—driven by cultural biases and implicit symbolic meanings in non-English text input—poses additional challenges. For example, when describing a person's appearance, Chinese prompts typically result in imagery featuring Asian characteristics and culturally specific contexts. In contrast, equivalent English descriptions predominantly yield imagery reflective of Western appearances. Such implicit cultural distributions and symbolic representations are challenging to capture through direct translation or fine-tuning approaches.

Existing approaches remain inadequate for complex semantic conditioning. Translation-based methods Zhang et al. (2022) lose culturally-specific meanings and symbolic expressions, deteriorating image quality and authenticity. Fine-tuning approaches Chen et al. (2022) that learn mappings between non-English and English representations struggle with culture-specific semantics unique to the source language, as these meanings often lack accurate English equivalents. Training entirely new models from scratch Feng et al. (2023b); Gu et al. (2022) requires massive computational resources and compromises compatibility with existing community-driven frameworks.

To address multilingual T2I challenges, we propose a novel approach that integrates a multilingual adaptation branch into existing model architecture while maintaining strong Flux community compatibility. Our method incorporates non-English embedding tokens into the cross-attention mechanism of MultiModal Diffusion Transformer (MMDiT) via a language adapter, enabling direct non-English linguistic control without altering the original architecture. Our approach introduces minimal additional parameters through a lightweight design that ensures efficiency and scalability. The backbone model remains fixed, allowing easy integration with existing Flux community plugins.

Moreover, to effectively address the semantic and cultural differences between languages, we employ a sophisticated two-stage training strategy. The model is trained using English and non-English data in the first stage. During this phase, we use a **Representation Alignment Loss** to align the features extracted by the non-English language encoder with the feature space of the original English language encoder. This alignment minimizes the linguistic feature domain gap and prevents the multilingual branch from needing to learn basic semantic concepts from scratch, accelerating the learning process. In the second stage, the model is fine-tuned exclusively on non-English data that contains culturally specific concepts. This allows the model to learn unique visual distributions associated with different cultures, enabling it to generate culturally accurate images and be visually faithful to the non-English prompts. Through this two-stage training, our approach accelerates convergence while enabling the model to capture both universal cross-lingual concepts and culture-specific semantics, thereby improving the quality and accuracy of generated images. Our core contributions can be summarized as follows :

- We propose the first method to natively adapt Flux for Chinese language support, enabling effective Chinese prompt handling.
- Our approach extends Flux to non-English environments while ensuring seamless compatibility with existing English-centric community plugins.
- Experimental results demonstrate consistent performance across various metrics and benchmarks, including general and culturally-specific image generation tasks.

## 2 RELATED WORK

### 2.1 DIFFUSION MODELS AND FLOW MATCHING

Diffusion models (DMs) Ho et al. (2020); Sohl-Dickstein et al. (2015) are powerful generative models that gradually generate images from Gaussian noise with a deep denoising model. Score-based generative models (SGMs) Song & Ermon (2019); Song et al. (2020b) learn to reverse a forward Ito process using stochastic differential equations (SDEs), training a score function $\nabla_{\boldsymbol{x}} \log p(\boldsymbol{x}, t)$ via denoising score matching (DSM) to guide sampling. Techniques like DDIM Song et al. (2020a),

ADM Dhariwal & Nichol (2021), CM Song et al. (2023), and latent diffusion Rombach et al. (2022) improve efficiency and output quality. Flow matching (FM) Lipman et al. (2023) and rectified flow Liu et al. (2022) offer deterministic alternatives by learning velocity fields over ODEs, enabling few-step or even single-step generation Yin et al. (2024). FM and SGMs are theoretically linked through Fokker-Planck equations and probability flow ODEs Song et al. (2020b); Evans (2010), inspiring hybrid models that balance diffusion's robustness with flow's efficiency Esser et al. (2024); Labs (2024); Labs et al. (2025).

## 2.2 ON THE ADAPTATION OF DIFFUSION GENERATIVE MODELS

Recent advances in generative foundation models have facilitated the generation of task-specific outputs without the need to train large backbone networks from scratch. A variety of adaptation methods have been proposed to support this capability. IP-Adapter Ye et al. (2023) adds lightweight cross-attention modules to inject reference images as prompts, enabling efficient style transfer and image-guided generation. ControlNet Zhang et al. (2023) attaches task-specific control branches (e.g., depth, pose, edges) to frozen DMs for precise structural control. Other parameter-efficient methods like LoRA and textual inversion Hu et al. (2022); Gal et al. (2022) reduce compute costs while maintaining quality. In flow matching (FM), models such as Flux. 1 and Flux Kontext Labs (2024) use deterministic flows for fast generation and editing, integrating well with adapter-based control, suggesting unified controllability across DMs and FM frameworks.

Multilingual adaptation is important for T2I generative models as it enables global accessibility, allowing users to use the model in their native languages. While previous works have made progress in supporting multilingual input, they often treated non-English languages as auxiliary. For example, Li et al. (2023) mitigated the language gap by translating English captions into other languages using a neural machine translation system. AltCLIP Chen et al. (2022) took a different route by enhancing diffusion models with the multilingual text encoder XLM-R. Meanwhile, ERNIE-ViLG 2.0 Feng et al. (2023a) trained a diffusion model using Chinese image-text pairs from scratch. However, these approaches did not effectively bridge native language and English-speaking communities. Liu et al. (2025) proposed the bridge diffusion, but it can only be applied to Unet-based models such as Stable Diffusion 1.5. In contrast, our work integrates native language models into the pretrained English-centric DiT, aiming to achieve a state-of-the-art multilingual generation framework.

## 3 PRELIMINARY

We denote $\boldsymbol{x}$ as the latent image embedding obtained from the VAE encoder. Let $\boldsymbol{\tau}_{EN}$ and $\boldsymbol{\tau}_{CN}$ represent the English and Chinese text embeddings $\boldsymbol{y}_{EN}$, $\boldsymbol{y}_{CN}$ after the **T5** Ni et al. (2021) and **Qwen** Yang et al. (2025) text encoders, respectively. We aim to train a model that generates images conditioned on a Chinese or an English text prompt. Formally, we strive to approximate the conditional distribution $p(\boldsymbol{x} \mid \boldsymbol{y})$, where we use $\boldsymbol{y}$ as a simplified notation for the text condition.

Under given text conditions $\boldsymbol{y}$, flow models, such as Flux.1 Labs (2024) utilize a velocity field $\boldsymbol{v}(\boldsymbol{x}, \boldsymbol{y}, t)$ to gradually turn noise $\boldsymbol{x}_0, \boldsymbol{\epsilon} \in \mathcal{N}(\boldsymbol{0}, \boldsymbol{I})$ at $t = 0$ into the data distribution $\boldsymbol{x}_1 \in p_{data}(\boldsymbol{x}|\boldsymbol{y})$ at $t = 1$. The dynamics of the marginal distribution $p(\boldsymbol{x}, t|y)$ are formulated by probability flow ODE and the continuity equations:

$$\begin{cases} \dfrac{\partial}{\partial t} p(\boldsymbol{x}, t|\boldsymbol{y}) = -\nabla_{\boldsymbol{x}} \cdot (\boldsymbol{v}(\boldsymbol{x}, \boldsymbol{y}, t) \cdot p(\boldsymbol{x}, t|\boldsymbol{y})), \\ \boldsymbol{x}_0 \sim p_0(\boldsymbol{x}|\boldsymbol{y}) = \mathcal{N}(\boldsymbol{0}, \boldsymbol{I}), \quad \boldsymbol{x}_1 \sim p_{data}(\boldsymbol{x}|\boldsymbol{y}). \end{cases} \tag{1}$$

In application, our adaptation of Flux to Chinese input follows a similar optimization strategy, with the primary difference being that the English embeddings $\boldsymbol{\tau}_{EN}$ are replaced by the Chinese embeddings $\boldsymbol{\tau}_{CN}$. To estimate the velocity field conditioned on a Chinese text prompt, we train a diffusion transformer to estimate the velocity field, denoted as $\boldsymbol{v}_{\boldsymbol{\theta}}(\boldsymbol{x}, \boldsymbol{y}, t)$, using a dataset consisting of images and Chinese captions: $\{\mathcal{X}_{train}, \mathcal{Y}_{train}\}$, via the flow matching loss:

$$\begin{cases} \mathcal{L}_{\boldsymbol{\theta}} = \mathbb{E}_{t, \boldsymbol{x}_i, \boldsymbol{\tau}_{CN}^i} \left[ \left\| \boldsymbol{v}_{\boldsymbol{\theta}}(\boldsymbol{x}, \boldsymbol{\tau}_{CN}^i, t) - (\boldsymbol{x}_i - \boldsymbol{\epsilon}) \right\|_2^2 \right] \\ \boldsymbol{x} = (1 - t) \cdot \boldsymbol{x}_i + t \cdot \boldsymbol{\epsilon}, \end{cases} \tag{2}$$

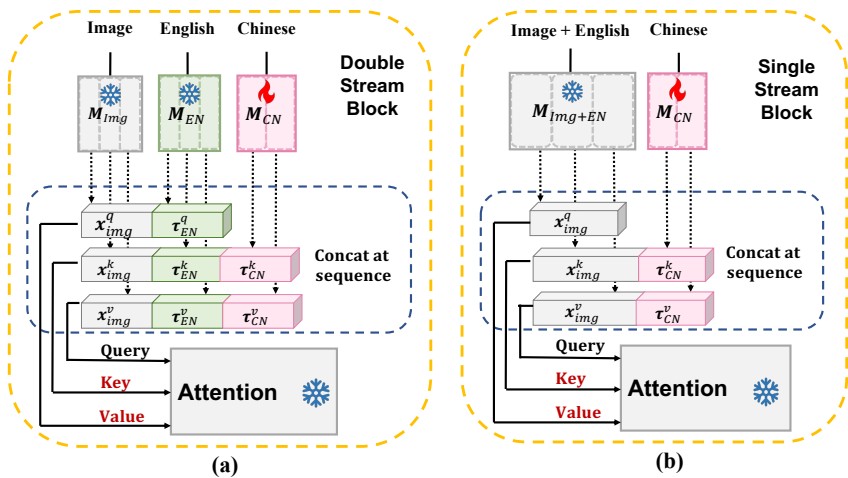

Figure 1: The overall pipeline of our proposed CTA-Flux. We only optimize the Chinese project layer in MMDiT blocks. The model is optimized with the flow matching generation loss (denoted as $\mathcal{L}_{GEN}$) and representation alignment loss (denoted as $\mathcal{L}_{RA}$).

Figure 2: The frameworks of our Chinese Linguistic Attention Branch (CLAB) in the MMDiT module, (a) and (b) shows the modification of the single stream and double stream block in each MMDiT module, respectively, where we add linear projection modules to map the Chinese embedding to Query, Key, and Value for the attention processor, denoted as $M_{CN}$.

where $t \sim \mathcal{U}(0,1)$, $x_i \sim \mathcal{X}_{train}$, $y_{CN}^i \sim \mathcal{Y}_{train}^{CN}$. The velocity field $v_\theta$ is estimated with a transformer such as DiT. During the training and inference, the English and Chinese text prompts are passed through the text encoder to obtain the text embeddings, denoted as $\tau_{EN}$ and $\tau_{CN}$.

## 4 METHOD

Our overall framework, illustrated in Figure 1, consists of two primary components: the visual backbone and the language branch. To maintain compatibility with community-developed plugins built on base generative models, we adopt the pretrained Flux diffusion transformer as our backbone. In addition, we employ the same pretrained pixel Encoder and Decoder modules as those used in Flux to provide image latent embeddings. On top of this foundation, we design a parallel language branch dedicated to processing Chinese text prompts. This branch delivers Chinese linguistic information, functioning either in conjunction with English inputs or on its own, to enhance the model's multilingual understanding and generation capabilities.

### 4.1 THE ARCHITECTURE OF THE LANGUAGE BRANCH

As illustrated in Figure 1, the multilingual text inputs are passed through the Chinese Linguistic Attention Branch (CLAB) Vaswani et al. (2023). Specifically, Chinese textual prompts are initially processed by a language-specific encoder capable of effectively capturing Chinese semantics. In our

framework, we adopt **Qwen 2.5** Yang et al. (2025) as the encoder for Chinese language inputs, while retaining **T5** Ni et al. (2021) as the encoder for English inputs, thereby providing language embedding tokens for conditional generation. These embeddings subsequently undergo dimensional alignment with the backbone branch's latent dimensions via a multi-layer perceptron (MLP)(Represented as the "English Adapter" and "Chinese Adapter" in Figure 1).

Given the intrinsic modality discrepancies between linguistic and visual domains, we employ a cross-attention mechanism rather than concatenation to integrate language information into the visual latents to facilitate the generation process. As illustrated in Figure 2, the aligned language tokens are projected into queries (conditionally updated based on task requirements), keys, and values (QKV) via dedicated projection layers at each layer within the MMDiT block of the backbone branch. The modified MMDiT blocks consist of a double-stream block (Figure 2 (a)) and a single-stream block (Figure 2 (b)). For the **double-stream block**, Image and English tokens are passed through their respective frozen QKV projection layers ($M_{img}$ and $M_{EN}$) to generate QKV representations (denoted as $x_{img}^q, x_{img}^k, x_{img}^v$ and $\tau_{EN}^q, \tau_{EN}^k, \tau_{EN}^v$) for the attention processor. In the **single-stream block**, Image and English input tokens are first concatenated and then passed through a frozen shared projection layer ($M_{img+EN}$) to produce mixed QKV representations (denoted as $x_{img+EN}^q, x_{img+EN}^k, x_{img+EN}^v$). For both double-stream and single-stream blocks, Chinese text embeddings are processed using a **trainable** Chinese projection layer $M_{CN}$ to generate their corresponding KV embeddings: $\tau_{CN}^k, \tau_{CN}^v$. These projected Chinese language embeddings interact with their image/English pairs in the MMDiT blocks through cross-attention operations.

This cross-attention strategy enables explicit and dynamic integration of non-English linguistic semantics into the backbone branch. Consequently, the CLAB provides nuanced, culturally-aware control signals during image synthesis, significantly enhancing the generation quality and semantic fidelity of outputs conditioned on non-English prompts. To ensure more effective training of the Language Branch and enable the model to learn control signals from non-English inputs accurately, we feed an empty string as English text inputs to the English language branch during training. This strategy prevents the model from relying on the pretrained Backbone's inherent ability to understand English prompts, forcing it to attend to the information provided by the Chinese language branch.

It is important to note that the English embedding projection layers in the Language Branch are frozen and not updated across the MMDiT blocks, and the non-English tokens are not used as queries in the cross-attention computation. This design stems from our observation that when Chinese branch tokens are used as queries in cross-attention, they inevitably attend to features from the English empty-text tokens in the Backbone Branch, which disrupts the Chinese control signals and results in semantically unstable or even random generations. By discarding the Chinese query tokens to prevent such interference, we maintain the effectiveness of Chinese text prompt conditioning throughout the generation process.

## 4.2 ADAPTATION STRATEGY

Our method preserves the architecture and parameters of the backbone branch entirely to ensure compatibility with existing community-developed plugins—such as ControlNet, LoRA, and custom checkpoints. Only the Chinese language branch parameters are updated during training, allowing our model to remain compatible with widely used tools and extensions within the Flux ecosystem.

Cross-lingual image generation presents two significant challenges: the linguistic feature distribution gap and visual feature distribution gap. The former arises from inherent linguistic differences across languages, and the latter stems from cultural biases embedded in the training data. To address these issues, we propose a carefully designed two-stage adaptation strategy. In the first stage, we mitigate the feature domain gap between the English and non-English language encoders by aligning their embedding spaces, enabling the Language Branch to capture shared semantics more effectively. In the second stage, we fine-tune the model exclusively on non-English data containing culturally specific concepts, thereby enabling the model to adapt to the target distribution shifts and generate images that reflect the unique cultural semantics of the source language.

In the first stage of training, we use a mixture of Chinese and English prompt texts. Each prompt—whether in Chinese or English—is passed through the Language Branch. We introduce an auxiliary supervision signal in addition to the standard image generation loss to accelerate the

learning process and address the cross-lingual adaptation challenge. This auxiliary objective mitigates the "linguistic feature distribution gap" between non-English and English encoders.

Concretely, we use each non-English prompt's semantically equivalent English counterpart solely to obtain features from the original English text encoder in the Backbone Branch. These English features are projected by the fixed MLP projection layer in the DiT backbone to match the DiT latent dimension, and are used only for computing the auxiliary alignment loss, without participating in the image generation process. Simultaneously, we obtain the projected non-English features from the Language Branch and minimize the mean squared error (MSE) between the projected non-English and corresponding English features. This auxiliary loss encourages the Language Branch to produce features compatible with the backbone's embedding space and facilitates effective multilingual adaptation. The formulation of this loss is given by:

$$\begin{cases} \mathcal{L}_p = MSE\left(\text{AvgPool}(\tau_{CN}), \text{AvgP}(\tau_{EN}^{aux})\right), \\ \mathcal{L}_{inter} = MSE\left(\tau_{CN}, \text{Intp}(\tau_{EN}^{aux}, \text{len}(\tau_{CN}))\right), \end{cases} \quad (3)$$

where $\tau_{EN}^{aux}$ represents the text embeddings derived from an English prompt semantically equivalent to the Chinese prompt employed to generate $\tau_{CN}$, obtained via English the text encoder, $MSE(\cdot)$ denotes the mean square error between two tensors, $\text{AvgP}(\cdot)$ denotes the average pooling on the sequence dimension of tokens, and $\text{Intp}(\cdot)$ interpolates the English feature to the length of the Chinese feature in the sequence dimension. The loss function consists of two parts: $\mathcal{L}_p$ and $\mathcal{L}_{inter}$. $\mathcal{L}_p$ is the Mean Squared Error (MSE) calculated on the feature after pooling the entire sentence, representing the overall alignment of the sentence-level representation. $\mathcal{L}_{inter}$ is the MSE between the text encoder's English tokens, which are interpolated to match the length of the Chinese tokens in the text encoder, aiming to align the features across the different token lengths.

In the first stage of training, we found that incorporating a threshold $\mathcal{D}_{threshold}$ for the auxiliary loss is necessary. If the auxiliary loss falls below this threshold, it is set to zero. The purpose of this approach is to avoid overemphasizing alignment, allowing the model to retain the subtle differences inherent in the language itself. This ensures that there is greater optimization potential for improving the generation quality. The alignment loss is formulated as:

$$\mathcal{L}_{RA} = \begin{cases} \mathcal{L}_p + L_{inter} & \text{if } \mathcal{L}_{RA} \geq \mathcal{D}_{threshold} \\ \mathcal{L}_p & \text{if } \mathcal{L}_{RA} < \mathcal{D}_{threshold}. \end{cases} \quad (4)$$

In the second stage of training, we use solely Chinese data and incorporate a wide range of concepts unique to Chinese culture, such as special holidays, cuisine, and traditional clothing. The training resolution is gradually increased, starting from 256, progressing to 512, and finally reaching 1024, to enable the model to capture culturally specific output targets. The overall training objective function $\mathcal{L}_{\boldsymbol{\theta}}$ is as follows:

$$\mathcal{L}_{\boldsymbol{\theta}} = \mathcal{L}_{GEN} + \mathcal{L}_{RA}, \quad (5)$$

where $\mathcal{L}_{GEN}$ follows the formulation in Eq. 2, and $\mathcal{L}_{RA}$ denotes the alignment loss.

### 4.3 INFERENCE STRATEGY

We can maintain consistency with the training phase during the inference phase by leaving the backbone text empty. Alternatively, we can input the same meaning in English into the backbone while inputting Chinese into the branch. We found that when Chinese is input into the branch while English is input into the backbone, it achieves better results than either using only the backbone or the branch independently.

## 5 EXPERIMENTS

### 5.1 EXPERIMENTAL SETUP

The training dataset consists of approximately one billion image-text pairs, primarily drawn from high-quality internal Chinese datasets, supplemented with publicly available English datasets. Roughly 60% of the data is Chinese and 40% is English. We ensure that Chinese data constitutes the majority to enhance the model's capacity to generate semantically rich Chinese content

Table 1: We compared CTA-Flux with other T2I models on the MS-COCO 256x256 dataset using FID-30K and the GenEval benchmark. **Bold** indicates the best performance (1st), and underline indicates the second-best performance (2nd).

| Model | Input Language | MS-COCO30K | | GenEval | | | | |
| --- | --- | --- | --- | --- | --- | --- | --- | --- |
| | | FID ↓ | CLIP Score ↑ | Single Obj. ↑ | Two Obj. ↑ | Counting ↑ | Color Attri. ↑ | Overall ↑ |
| LDM Rombach et al. (2022) | English | 37.45 | 25.84 | 0.92 | 0.29 | 0.23 | 0.06 | 0.37 |
| SD1.5 Rombach et al. (2022) | English | 22.87 | 27.09 | 0.97 | 0.38 | 0.35 | 0.05 | 0.43 |
| SDXL Podell et al. (2023) | English | 18.64 | **28.47** | 0.98 | 0.74 | 0.39 | 0.23 | 0.55 |
| Flux Labs (2024) | English | 16.39 | 27.94 | 0.98 | **0.81** | **0.69** | 0.47 | **0.65** |
| BDM Liu et al. (2025) | Chinese and English | 28.34 | 23.66 | 0.97 | 0.32 | 0.31 | 0.08 | 0.41 |
| CTA-Flux (Ours) | Chinese and English | **15.40** | 27.98 | **0.99** | 0.74 | 0.53 | **0.50** | 0.62 |

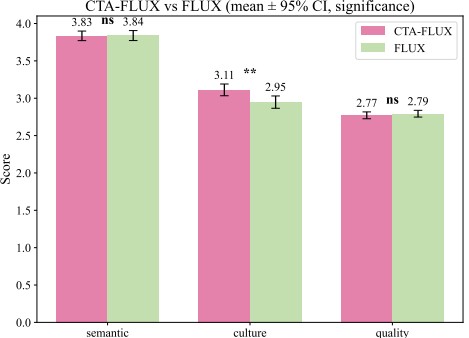

Figure 3: Culturally Human Eval Results

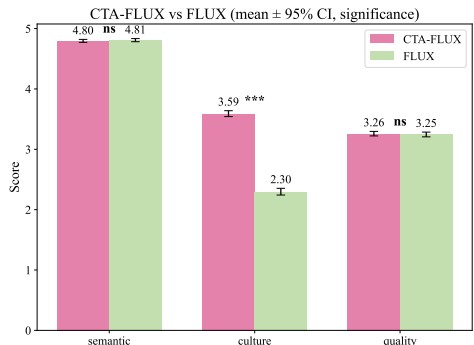

Figure 4: Cultural-Neutral Eval Results

and reduce the dominance of English-centric concepts. Before training, rigorous filtering is applied to exclude samples with watermarks, low aesthetic quality, or weak image-text alignment We adopt FLUX.1-dev Labs (2024) as the base generative model. For the Chinese text encoder, we use Qwen2.5-VL-7B-Instruct Bai et al. (2025), and for the English text encoder, we adopt T5 Ni et al. (2021), with the framework consistent with FLUX.1-dev, and the weights of the T5 text encoder are frozen during training. The hyperparameter $\mathcal{D}_{\text{threshold}}$ is set to 0.05.

The training of the diffusion transformer is built upon Flux's latent space, which employs a Variational Autoencoder (VAE) to transform images between pixel space and latent representations. The training pipeline is implemented in PyTorch-Diffusers von Platen et al. (2022). We use the AdamW optimizer Kingma & Ba (2017); Loshchilov & Hutter (2019) with a learning rate of 1e-5 and a total batch size of 3200. Training is conducted over two weeks on 32 NVIDIA A800 GPUs.

## 5.2 QUANTIVATE EVALUATION

**Evaluation on COCO and GenEval Benchmark** Following prior work Rombach et al. (2022); Podell et al. (2023); Labs (2024), we evaluate CTA-Flux on the COCO Lin et al. (2015) 256×256 dataset using the zero-shot Fréchet Inception Distance (FID). To ensure fairness, we adopt the standardized MS-COCO 30K benchmark with Recaption-COCO Li et al. (2024), which provides more diverse and balanced captions than the original short COCO annotations. In addition, we report results on GenEval Ghosh et al. (2023), which measures fine-grained compositional aspects such as object count, color, and attribute consistency, offering complementary insights into generation quality.

As shown in Table 1, CTA-Flux achieves a lower FID (15.40) and a slightly higher CLIP score (27.98) compared to Flux, indicating that the introduction of Chinese capability does not compromise, but rather marginally enhances, the original performance of Flux.

On GenEval, CTA-Flux (0.62) performs on par with Flux (0.65) and clearly surpasses BDM (0.41), showing that the added Chinese capability does not weaken compositional reasoning.

**Human Eval Results** We built a human evaluation benchmark with 200 culturally specific and 200 culturally neutral prompts generated by GPT-4o, spanning 15 categories (e.g., characters, festivals, daily-life). For each prompt, CTA-Flux and FLUX produced four images, which were rated on semantic consistency, cultural authenticity, and image quality using a 1–5 Likert scale (see Appendix for criteria).

Table 2: We test the CLIP score ↑ to measure the Chinese cultural inclination.

| Class | Human | | Architecture | | Clothing | |
| Culture | Chinese | Caucasian | Chinese | Caucasian | Chinese | Caucasian |
|---|---|---|---|---|---|---|
| Flux Labs (2024) | 17.099 | **18.953** | 20.740 | **24.346** | 21.292 | **23.286** |
| CTA-Flux(ours) | **19.849** | 18.595 | **24.635** | 24.104 | **25.736** | 22.995 |

| Class | Food | | Festival | | Art | |
| Culture | Chinese | Caucasian | Chinese | Caucasian | Chinese | Caucasian |
|---|---|---|---|---|---|---|
| Flux Labs (2024) | 19.265 | **22.168** | 21.861 | **23.826** | 21.162 | **22.088** |
| CTA-Flux(ours) | **22.869** | 21.220 | **25.472** | 22.812 | **25.889** | 21.503 |

As shown in Fig. 3, our model (CTA-Flux) achieves comparable performance to FLUX in terms of semantic consistency (3.83 vs. 3.84, $p = 0.94$, not significant) and visual quality (2.77 vs. 2.79, $p = 0.005$, not significant), indicating that the proposed method does not compromise the fidelity of content or aesthetic appeal. Importantly, CTA-Flux significantly improves cultural authenticity (3.11 vs. 2.95, $p = 0.51$), highlighting its effectiveness in enhancing cultural representation without sacrificing semantic or perceptual quality. Statistical significance is assessed using independent two-sample $t$-tests, where the $p$-value denotes the probability of observing the result under the null hypothesis; $p < 0.05$, $p < 0.01$, and $p < 0.001$ are denoted by "*", "**", and "***", respectively, while "ns" indicates no significant difference.

For culture-neutral prompts, as reported in 4, CTA-Flux shows comparable performance to FLUX in semantic consistency (4.80 vs. 4.81, $p = 0.50$, ns) and image quality (3.26 vs. 3.25, $p = 0.63$, ns), while achieving a substantial gain in cultural authenticity (3.59 vs. 2.30, $p < 0.001$), further confirming its advantage in capturing Chinese cultural semantics.

**Chinese Cultural Tendency** To evaluate the cultural inclination of the model, we use GPT-4o to generate 150 culture-neutral prompts in four categories: people, architecture, cuisine, festivals, clothing and art. We inference with these prompts and then compute the CLIP similarity between generated images and textual concepts "Chinese category" and "Caucasian category". Higher similarity indicates the model's stronger inclination toward the corresponding cultural concept

As shown in Table 2, CTA-Flux consistently achieves higher CLIP scores on Chinese-related categories while maintaining comparable performance on Caucasian ones, indicating a clear shift towards stronger Chinese cultural inclination.

### 5.3 QUALITATIVE RESULTS

**Native Language Semantics** We validated the Chinese semantics, and the experimental results are shown in Figure 5. From the figure, it is evident that Flux Labs et al. (2025) faces the issue of ambiguity between Chinese and English. Additionally, Flux lacks cultural bias, whereas our CTA-Flux exhibits a stronger inclination towards Chinese culture. While BDMv1.0 Liu et al. (2025) supports Chinese prompt input and demonstrates a degree of cultural alignment with Chinese semantics, our CTA-Flux achieves superior performance in terms of image generation quality and aesthetic fidelity. Both our CTA-Flux and Seedream Gao et al. (2025) demonstrate significant cultural bias, but compared to Seedream Gao et al. (2025), CTA-Flux is more compatible with various plugins in the open-source Flux community, offering better community support.

**Community Support** To assess the extent of compatibility with the Flux community, we tested our model using popular LoRA weights released by the community. The test results, as shown in Figure 6, demonstrate that our model is compatible with various styles of LoRA, exhibiting community.

### 5.4 ABLATION STUDY

**Impact of Loss Function on Performance** To investigate the impact of adding loss and threshold on model performance, we conducted experiments under the same training conditions with 400,000 samples. We compared the model's generation quality using COCO FID Heusel et al. (2018) and CLIP score Radford et al. (2021) under different configurations: no loss, no threshold; no loss with threshold; and with loss but without threshold. The results in table 3 demonstrate that adding loss improves the model's semantic understanding and generation quality. Incorporating the threshold further enhances generation quality while maintaining strong semantic understanding.

Table 3: We conducted tests with the same number of training steps for the ablation study on adding the loss function. The FID was evaluated using a fixed set of 3000 images selected from the COCO dataset as the test samples.

| Loss function | | | | Metrics | |
|---|---|---|---|---|---|
| $\mathcal{L}_{RA}$ | $\mathcal{L}_p$ | $\mathcal{L}_{inter}$ | $D_{threshold}$ | FID ↓ | CLIP score ↑ |
| ✗ | - | - | - | 93.06 | 30.47 |
| | ✓ | ✗ | ✗ | 77.10 | 31.42 |
| ✓ | ✓ | ✓ | ✗ | 58.86 | 31.28 |
| | ✓ | ✓ | ✓ | **53.52** | **31.40** |

Table 4: The ablation study on whether the Chinese embedding feature should be updated as a query was conducted.

| Chinese embedded token | FID ↓ | CLIP score ↑ |
|---|---|---|
| w/ update | 277.00 | 30.47 |
| w/o update | **81.58** | **31.42** |

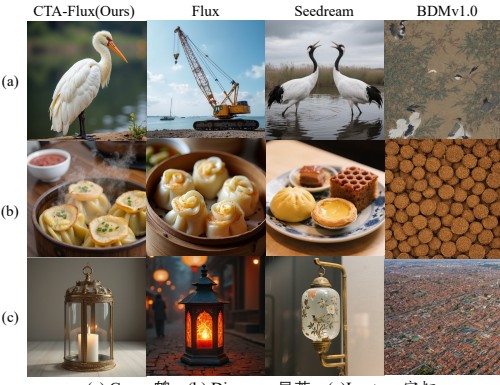

(a) Crane.鹤。(b) Dim sum.早茶。(c)Lantern.宫灯。

Figure 5: The performance of different T2I models in generating descriptions related to Chinese culture is compared. Our CTA-Flux not only maintains the generation quality of Flux but also exhibits a strong inclination towards Chinese culture and offers excellent community compatibility.

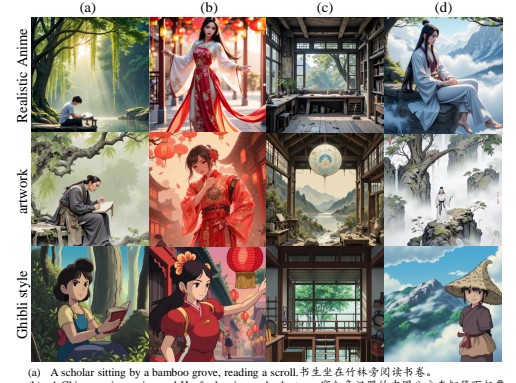

(a) A scholar sitting by a bamboo grove, reading a scroll.书生坐在竹林旁阅读书卷。
(b) A Chinese princess in a red Hanfu dancing under lanterns.穿红色汉服的中国公主在灯笼下起舞。
(c) A tea house with carved wooden screens and bamboo chairs.雕花木窗与竹椅围绕的传统茶馆。
(d) A wise Taoist monk meditating on a misty mountain top.一位智慧的道士在雾气缭绕的山顶冥想。

Figure 6: By using CTA-Flux with different LoRA modules, images generated with various LoRA prompts demonstrate CTA-Flux's strong community adaptability. This framework can generate high-quality, aesthetically pleasing, and high-resolution images.

**Avoiding the Use of Chinese Embedding as Query in Cross-Attention** In T2I generation models, text embeddings are typically used as queries in the cross-attention operation to facilitate information fusion across modalities. We conducted an ablation experiment to explore whether the Chinese text embeddings should be used as queries in the cross-attention computation and update their own features. We tested a model trained with 160,000 samples using COCO 256x256 FID and CLIP score metrics. The experimental results show that allowing the Chinese branch tokens to update during the cross-attention operation severely degrades the model's generation quality and semantic understanding. We speculate that this may be due to the strong random conditions introduced by the empty text input in the English branch during training, which, when the Chinese tokens query the English tokens, makes them highly susceptible to these random conditions, weakening the representation capability of the Chinese tokens.

## 6 CONCLUSION

Our work realizes the adaptation of the Flux model to Chinese text by training a lightweight Chinese language branch within the MMDiT blocks, avoiding the training of the entire T2I model from scratch. The English and image input branches remain frozen during training and inference, preserving compatibility with the base Flux model and the broader English T2I community. Through qualitative visualizations and quantitative evaluations on multiple benchmarks, we demonstrate the effectiveness of our method in understanding Chinese text and generating semantic-related contents. Moreover, our approach is not limited to Chinese—it is promising for multilingual T2I adaptation tasks, highlighting the generalization ability of the proposed framework.

ETHICS STATEMENT

This work adheres to the ICLR Code of Ethics.[1] Our research does not involve human subjects, sensitive personal data, or identifiable private information. The datasets used in our experiments consist of two parts: (1) publicly available datasets that are widely used in the literature, with all usage in compliance with their respective licenses, and (2) internal datasets that are also used in accordance with proper licensing. We do not foresee direct harmful use of our methodology, but we acknowledge that, as with many machine learning techniques, there exists a potential for misuse in unintended application domains. We encourage responsible usage and further discussions within the community regarding broader social and ethical impacts.

REPRODUCIBILITY STATEMENT

We have taken several steps to ensure reproducibility of our results. All model architectures, training settings, and evaluation protocols are described in detail in Section 4.3. We will release all code and trained models in the future to enable reproduction of our main experiments and to support further research based on our work.

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

## LLM USAGE STATEMENT

In accordance with the ICLR policy on the use of large language models (LLMs), we disclose the role of LLMs in the preparation of this work. LLMs were used solely for translation (Chinese to English) and for language polishing to improve readability and grammar of the manuscript. They were not involved in ideation, dataset creation, model design, or experiment execution. We emphasize that the technical contributions, including the Representation Alignment Loss and the non-English branch are entirely part of the research methodology and do not constitute LLM-assisted writing. The authors take full responsibility for the correctness and originality of the content presented in this paper.

## A    MORE QUANTIVATE EVALUATION

To more comprehensively assess how our CTA-Flux model interprets complex concepts, fine-grained semantics, and structured instructions, we adopt T2I-CompBench++ Huang et al. (2025) and DPG-Bench Hu et al. (2024). T2I-CompBench++ is a compositional reasoning benchmark that examines whether a model can correctly bind attributes, resolve spatial and non-spatial relations, understand numeracy, and follow multi-object descriptions. With roughly 8k category-specific prompts covering attribute binding (color, shape, texture), object relations, numeracy tasks, and higher-level compositions, it provides a more fine-grained view of semantic consistency under bilingual prompting.

DPG-Bench (Dense Prompt Graph Benchmark) complements this by evaluating a model's ability to follow long, densely structured, graph-like instructions. Its 1,065 prompts contain rich entity dependencies and multi-step semantic constraints, allowing us to more thoroughly inspect whether a text-to-image model maintains stable semantic grounding when handling complex, cross-token or multi-hop instructions—an aspect particularly relevant when extending the model to Chinese understanding.

Table 5: **Performance Comparison on T2I-CompBench++ Huang et al. (2025).** Best results in each column are **bolded**, and second best values are underlined.

| Method | Input Lang. | Attribute Binding | | | Object Relationship | | | Numeracy↑ | Complex↑ |
|---|---|---|---|---|---|---|---|---|---|
| | | Color↑ | Shape↑ | Texture↑ | 2D-Spatial↑ | 3D-Spatial↑ | Non-Spatial↑ | | |
| SD1.5 Rombach et al. (2022) | English | 37.58 | 37.13 | 41.86 | 11.65 | – | 31.12 | – | 30.47 |
| SDXL Podell et al. (2023) | English | 58.79 | 46.87 | 52.99 | 21.31 | 35.66 | **31.19** | 49.91 | 32.37 |
| Flux Labs (2024) | English | 73.79 | **51.96** | **64.64** | **28.50** | **41.62** | 30.72 | **62.46** | **36.56** |
| CTA-Flux (Ours) | Chinese & English | **76.70** | 50.24 | 63.44 | 25.16 | 39.22 | 30.79 | 56.50 | 35.17 |

Table 6: **Performances on DPG-Bench Hu et al. (2024).** Best scores in each column are **bolded**, and the second best scores are underlined.

| Method | Input Lang. | Global | Entity | Attribute | Relation | Other | Overall↑ |
|---|---|---|---|---|---|---|---|
| SDv1.5 Rombach et al. (2022) | English | 74.63 | 74.23 | 75.39 | 73.49 | 67.81 | 63.18 |
| PixArt-$\alpha$ Chen et al. (2023) | English | 74.97 | 79.32 | 78.60 | 82.57 | 76.96 | 71.11 |
| SDXL Podell et al. (2023) | English | 83.27 | 82.43 | 80.91 | 86.76 | 80.41 | 74.65 |
| Flux Labs et al. (2025) | English | **89.51** | 85.81 | **88.31** | **91.28** | **90.11** | **83.24** |
| CTA-Flux (Ours) | Chinese and English | 88.36 | **87.64** | 86.36 | 88.52 | 87.22 | 80.23 |

Table 5 summarizes the results on T2I-CompBench++. Flux delivers the strongest overall performance, achieving the best scores in most dimensions, especially in Shape, Texture, Spatial reasoning, Numeracy, and Complex prompts. Despite handling the more challenging Chinese and English mixed-language setting, CTA-Flux remains highly competitive, with performance very close to Flux and even surpassing it on Color (76.70 vs. 73.79). This shows that adding Chinese understanding does not degrade compositional reasoning ability—instead, our bilingual alignment preserves the strengths of Flux while improving certain aspects of fine-grained attribute binding.

Table 6 reports results on DPG-Bench. Flux achieves the best overall performance, obtaining the highest scores across nearly all dimensions, including Global, Attribute, Relation, and Other reasoning. Under the more challenging Chinese and English mixed-language setting, CTA-Flux remains

highly competitive, with scores that closely match Flux and even achieve the best result on the Entity dimension (87.64 vs. 85.81).

## B  DETAILS OF DATASETS

In the first stage of training, we utilized a large-scale dataset consisting of mixed Chinese and English captions. The distribution of the "English", "Chinese-long", and "Chinese short" captions is illustrated in Fig. 7. To ensure robust performance across captions of different lengths, we maintained a balanced proportion between short and long Chinese prompts during training. This design encourages the model to generalize well under diverse linguistic conditions, including semantic granularity and sentence complexities.

The construction of the dataset used for the first-stage training is illustrated in Fig. 7. We collected approximately 150 million images, each paired with both Chinese and English captions, by using existing image-text datasets and generating additional captions using a recaptioning model. Specifically, we first collected image-caption pairs from the LAION Schuhmann et al. (2022) and GRIT Fan et al. (2025) datasets as the visual input for training, and supplemented the captions using CogVLM2 Hong et al. (2024) and Mantis Jiang et al. (2024). To further supplement the training set, we included images generated by Flux-dev Labs (2024), along with their corresponding captions produced by captioning models.

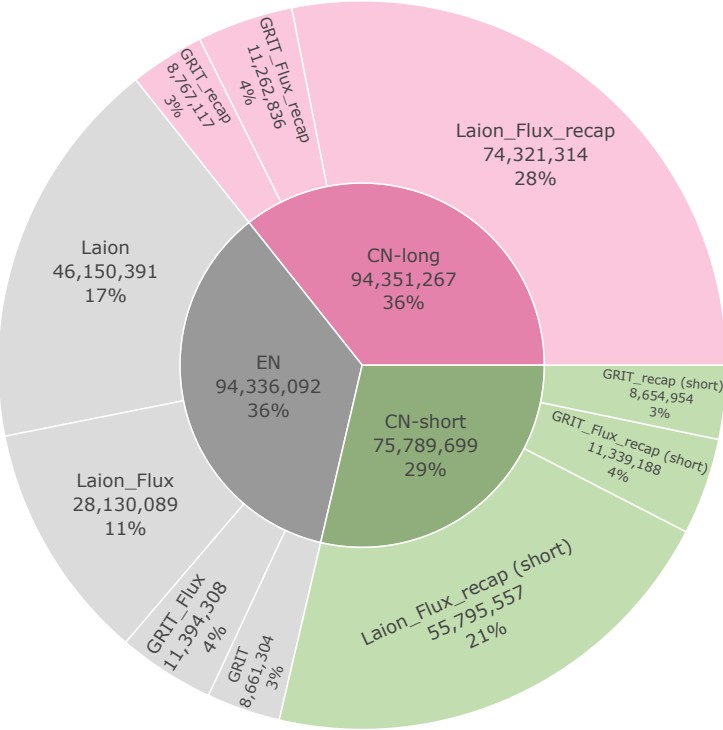

Figure 7: Caption language distribution across different datasets used in Stage 1 training. The root categories represent the caption language types: **EN** (English), **CN-long** (long-form Chinese), and **CN-short** (short-form Chinese). Subcategories denote the data sources or processing steps: **GRIT** and **Laion** are public datasets where the captions were originally collected; The suffix **Flux** indicates that the associated image was re-generated using Flux-dev; The suffix **recap** denotes that the caption has been rewritten or adjusted during preprocessing.

In the second-stage fine-tuning phase, we aimed to optimize the model for better alignment with the data distribution of Chinese culture. To achieve this, we curated a dataset of 40,000 real-world images paired with Chinese captions. Each pair was manually collected and annotated to ensure the inclusion of culture-specific visual and semantic concepts relevant to Chinese culture. We then

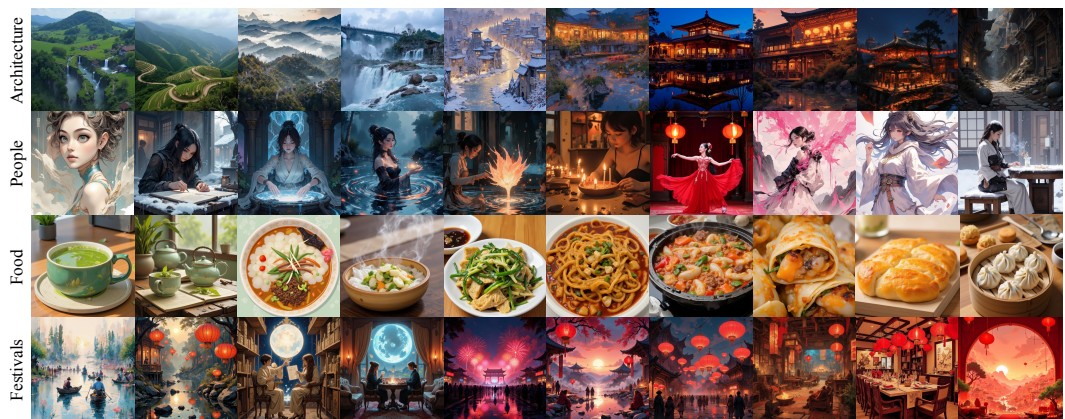

Figure 8: Selected samples generated by our CTA-Flux model using Chinese text prompts and various English plugins.

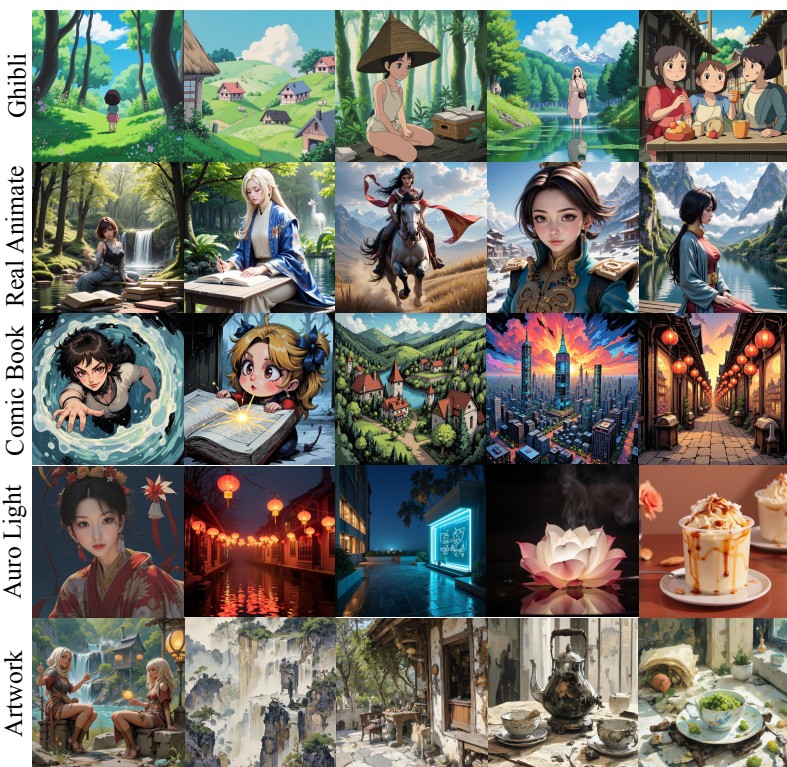

Figure 9: Additional generation results of CTA-Flux guided by various community LoRA styles. The model is compatible with diverse stylistic prompts while preserving high image fidelity and aesthetic quality.

fine-tuned the weights of the first-stage model on this curated dataset to enhance its understanding of Chinese textual input and improve its ability to generate culturally grounded images.

# C  MORE QUALITATIVE RESULTS

## C.1  SELECTED SAMPLES

To further demonstrate the effectiveness of our proposed model, we present a diverse set of high-quality images generated from Chinese prompts across four thematic categories: architecture, people, food, and festivals, corresponding to the first through fourth rows, respectively, as illustrated in Fig. 8. The generated samples show that our CTA-Flux model produces visually appealing and high-fidelity images and faithfully captures the language-specific semantics of the Chinese prompts, reflecting rich Chinese cultural elements in each image.

## C.2  MORE PLUGIN COMPATIBILITY RESULTS

To further evaluate the compatibility of our CTA-Flux model with various widely used community plugins, we conducted a series of experiments using additional Style-LoRA plugins available within the Flux community. As illustrated in Fig. 9, each row shows images generated in a specific LoRA style, conditioned on either Chinese or English text input. This demonstrates the model's adaptability to diverse stylistic controls across different languages. The generated images exhibit high visual quality and aesthetic property, confirming the effectiveness of CTA-Flux in supporting various LoRA-guided generation settings.

## C.3  VISUALIZATION OF CULTURE-SPECIFIC EXPERIMENT

Fig. 10 presents qualitative comparisons from the Culture-Specific experiment. While CTA-Flux produces images with visual fidelity comparable to Flux, it demonstrates noticeably stronger understanding of Chinese cultural concepts, including traditional garments, food, festivals, and architectural. This improvement stems from CTA-Flux's ability to internalize Chinese-specific semantics rather than relying on indirect English translations. As a result, the model mitigates the inherent bias of English-centric text-to-image systems and achieves more precise grounding under native Chinese prompts.

## C.4  VISUALIZATION OF CULTURE-NEUTRAL EXPERIMENT

Fig. 11 shows qualitative examples from the Culture-Neutral experiment. When the prompt involves human-related or implicitly culture-dependent descriptions, CTA-Flux naturally leans toward East-Asian appearances or stylistic cues, reflecting its enhanced grounding in Chinese cultural semantics. However, when the prompt is entirely culture-agnostic—such as natural landscapes or object-centric scenes—CTA-Flux produces images with quality and composition closely aligned with Flux, without introducing unnecessary Chinese elements or artifacts. These results indicate that CTA-Flux maintains cultural appropriateness: it accurately reflects cultural priors when relevant, while refraining from over-injecting culture-specific features when the prompt does not call for them.

## C.5  VISUALIZATION OF CULTURE-METAPHORICAL DESCRIPTIONS

To further examine the cultural semantic capabilities of our model, we visualize two complementary aspects (Figs.12 and 13).

First, our model shows a strong ability to interpret complex cultural metaphors, including idioms, poetic imagery, and figurative expressions that require understanding beyond literal word meanings. As shown in Fig.12, the idiom "cars, water. horses, dragons" describes a highly dynamic and bustling traffic flow in Chinese. When directly translated into English, however, Flux misinterprets the phrase as a set of unrelated objects ("car," "water," "horse," "dragon"), leading to compositional errors. In contrast, our method correctly grasps the intended metaphorical sense and produces a coherent visual depiction of a crowded urban traffic scene, demonstrating its superior capability in modeling culturally anchored semantics.

Second, we evaluate whether the model introduces cultural stereotypes or performs naive visual collage when the prompt contains no explicit cultural cues. As illustrated in Fig. 13, our model generates visually rich and realistic scenes—featuring fine-grained garment textures, detailed shoe

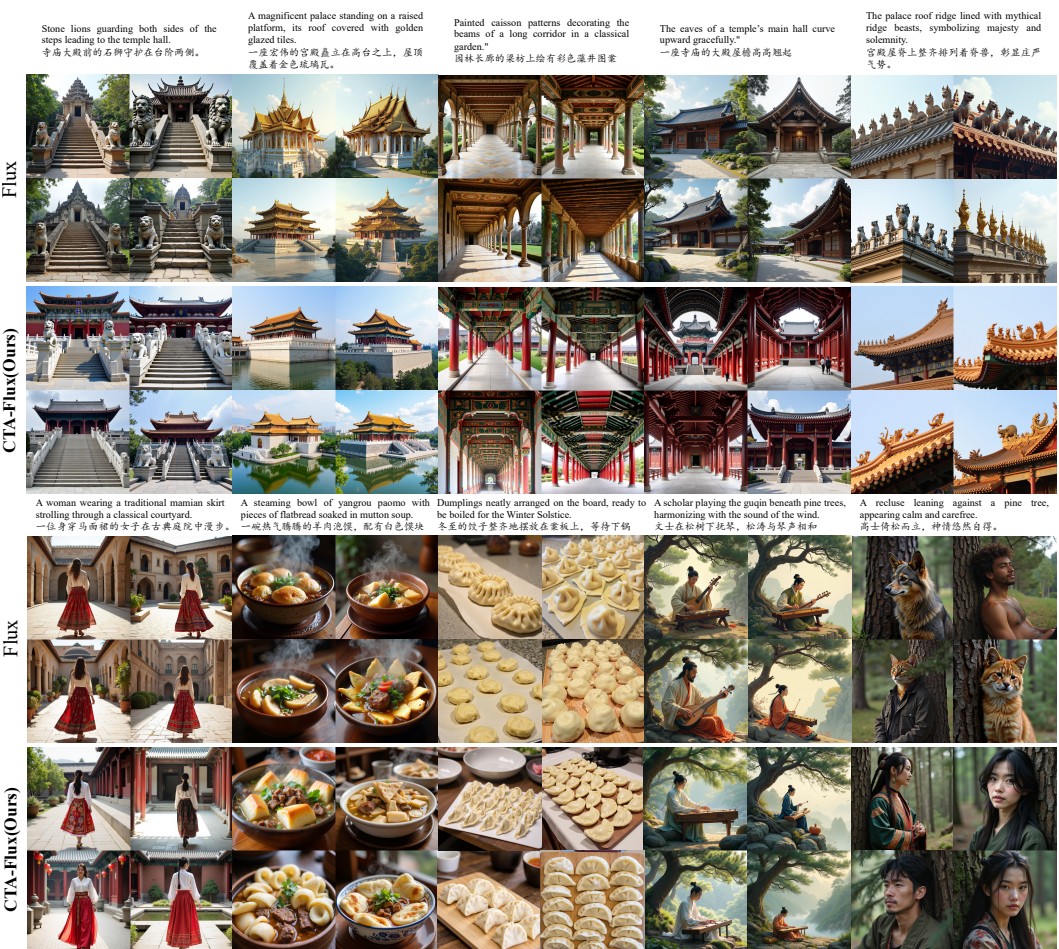

Figure 10: **Qualitative results on the Culture-Specific experiment.** All visualizations are randomly sampled four times for each prompt to illustrate the model's consistency and robustness. CTA-Flux demonstrates high-fidelity generation quality comparable to Flux, while providing significantly more accurate interpretations of Chinese cultural concepts, such as traditional clothing, regional cuisine, and architectural styles. The model effectively captures fine-grained cultural cues expressed in Chinese prompts and reduces the bias of English-centric models when handling culture-dependent semantics.

soles, and the reflective wetness of the pavement—without injecting unwarranted Chinese cultural symbols or assembling irrelevant visual components. The generated characters naturally appear as Asian individuals, consistent with the scene's visual context, but their clothing spans modern and diverse styles, such as business suits or everyday dresses, rather than being uniformly replaced with traditional Chinese garments. Moreover, the urban background adopts a Shanghai-like skyline, offering an East-Asian ambience that emerges organically from the image context rather than from oversimplified cultural tokenization. These results demonstrate that our model avoids culturally stereotypical or collage-like generation behavior, instead producing coherent, contextually grounded, and high-fidelity imagery even when cultural semantics are not explicitly provided.

## C.6 ZERO-SHOT MULTILINGUAL GENERALIZATION

Although our model is trained exclusively with Chinese and English text supervision, we further investigate its potential for multilingual generalization by evaluating prompts in four additional languages: Spanish, French, Russian, and Italian. As shown in Fig. 11, the model consistently produces semantically plausible images that reflect the core meaning of the input prompts, despite never being exposed to these languages during training. Since the Qwen text encoder inherently supports

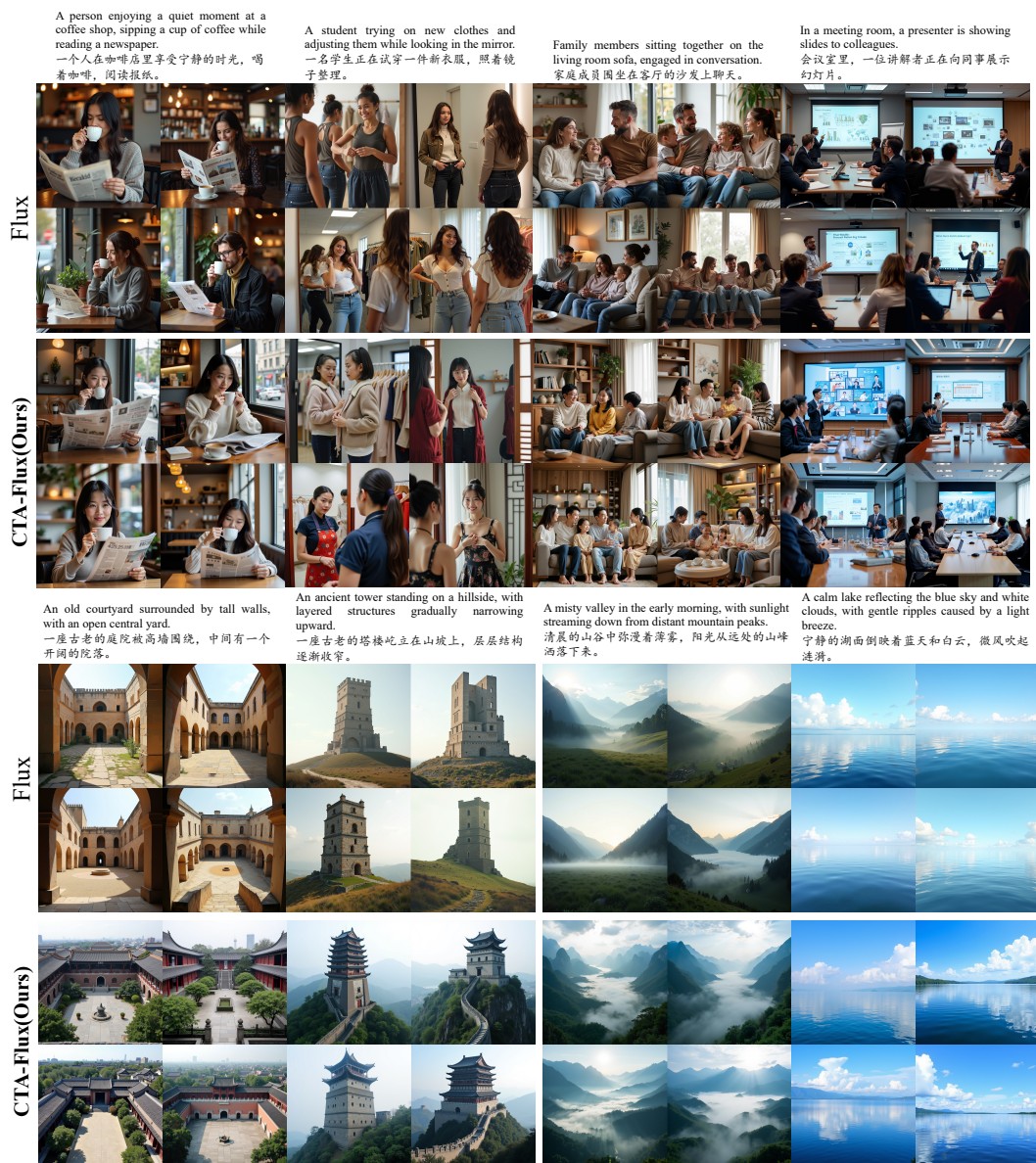

Figure 11: **Qualitative results on the Culture-Neutral experiment.** All visualizations are randomly sampled four times for each prompt to illustrate the model's consistency and robustness. For prompts without explicit cultural references, CTA-Flux exhibits a mild but coherent preference toward East-Asian cultural representations in human-related descriptions (e.g., facial appearance or traditional aesthetics), while maintaining high visual fidelity comparable to Flux. For prompts unrelated to human or cultural semantics (e.g., natural scenes), CTA-Flux generates images that match Flux in both quality and content without artificially inserting Chinese-specific elements.

a wide range of languages, our alignment strategy—focused only on bridging Chinese and English—can serve as an effective conduit for transferring its latent multilingual capability into the generation process. When the alignment is sufficiently robust, the model can therefore exhibit zero-shot generalization to languages it has never seen during training. This observation indicates that our Stage-1 semantic alignment strategy successfully anchors the model to a language-agnostic conceptual space, enabling it to transfer semantic understanding across unseen linguistic domains. These results highlight the broader extensibility of our method and suggest that lightweight cross-lingual adaptation can be obtained even without explicit multilingual supervision.

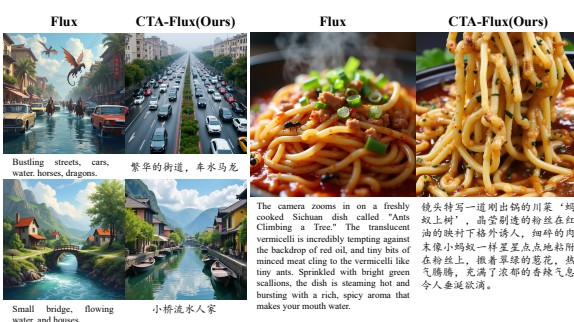 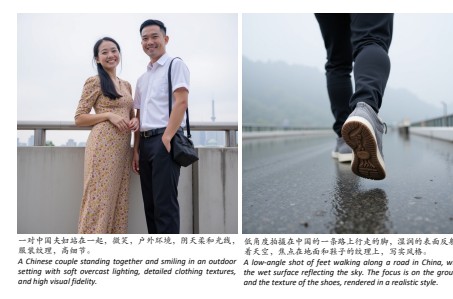

Figure 12: **Understanding Complex Cultural Metaphors.** Our model demonstrates strong capability in interpreting culturally grounded metaphorical expressions—including idioms, proverbs, and poetic references—rather than performing literal keyword matching or object-wise collage.

Figure 13: **Culture-Neutral Generation without Stereotypical Artifacts.** When prompts contain no explicit cultural description, our model generates high-fidelity images with fine-grained textures—such as clothing patterns, shoe details, and road reflections—while avoiding the injection of unwarranted cultural stereotypes.

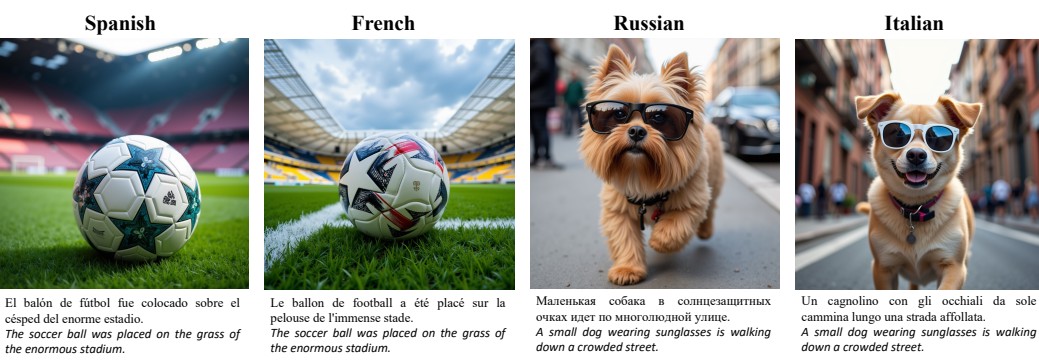

Figure 14: **Visualization of Multi-Language Generalization.** We qualitatively evaluate the cross-lingual robustness of our model by feeding prompts in four unseen languages—Spanish, French, Russian, and Italian—although the model is trained only with Chinese–English data. The results show that our approach can correctly interpret the essential semantics of prompts from these additional languages and generate reasonable images, demonstrating strong potential for multilingual generalization without any explicit training on these languages.

# D HUMAN EVALUATION CRITERIA

We conduct two human evaluation studies to measure how well the models understand Chinese prompts and represent Chinese cultural semantics. Each study uses a different set of prompts and evaluation dimensions, tailored to the cultural specificity of the input. The overall results of these two studies are visualized in Fig. 3 (culturally specific prompts) and Fig. 4 (culture-neutral prompts), while the detailed annotation criteria are defined as follows.

**Culture-Specific Experiment.** This experiment uses prompts that explicitly describe Chinese cultural concepts (e.g., traditional clothing, festivals, architecture). Annotators evaluate the following three dimensions:

**1. Semantic Consistency (1–5).** Measures whether the image correctly reflects the objects, relations, and attributes described in the Chinese prompt.

- **1 – Completely Inconsistent:** Image content is largely unrelated to the prompt; major objects incorrect or missing.
- **2 – Poor:** Some relevant elements appear, but core objects or relations are incorrect.
- **3 – Fair:** Main objects present but relations or attributes have noticeable errors.

- **4 – Good:** Image mostly matches the prompt with minor semantic inaccuracies.
- **5 – Very Consistent:** Objects, relations, and attributes fully align with the prompt.

**2. Cultural Authenticity (1–5).** Evaluates whether the Chinese cultural elements are represented correctly and faithfully, avoiding fabricated, stereotypical, or historically inaccurate patterns.

- **1 – Completely Incorrect:** No authentic Chinese elements, or elements are factually wrong or stereotypical.
- **2 – Mostly Incorrect:** Some cultural elements appear, but are inaccurate, misleading, or historically incorrect.
- **3 – Partially Correct:** Recognizable Chinese elements exist, but with noticeable cultural inaccuracies or mixed motifs.
- **4 – Mostly Correct:** Cultural elements generally accurate, with minor discrepancies.
- **5 – Highly Authentic:** Accurate and faithful representation of Chinese traditions with correct stylistic and cultural details.

**3. Image Quality (1–5).** Assesses visual fidelity, coherence, and aesthetic quality.

- **1 – Very Poor:** Severe artifacts or structural failures.
- **2 – Poor:** Noticeable artifacts or incoherent object structures.
- **3 – Fair:** Acceptable quality with minor flaws.
- **4 – Good:** Clear, coherent, and visually pleasing.
- **5 – Excellent:** High aesthetic quality with clean details and coherent composition.

**Culture-Neutral Prompts.** This experiment uses prompts that do not contain explicit Chinese cultural elements. Annotators evaluate three aspects of each generated image:

**1. Semantic Consistency (1–5).** (Same definition as in the Culture-Specific Experiment; see metric D.)

**2. Cultural Inclination (1–5).** Evaluates whether the generated image presents a more Western or Eastern (Chinese) cultural style.

- **1 – Strongly Western:** The image predominantly exhibits clear Western visual elements, such as Western facial features, clothing, architecture, cuisine, or artistic styles, with no noticeable traces of Eastern cultural influence.
- **2 – Western-Oriented:** The image is mainly Western in appearance, though a small number of neutral or culturally ambiguous elements may be present.
- **3 – Neutral / Mixed:** The image shows no obvious cultural leaning, or contains a mixture of Western and Eastern elements without either side being dominant.
- **4 – Eastern-Oriented:** The image primarily reflects Chinese/Eastern cultural elements, though minor neutral or Western features may still appear.
- **5 – Strongly Eastern:** The image strongly embodies Chinese/Eastern cultural characteristics, with individuals, scenes, clothing, architectural motifs, and other visual elements highly aligned with an Eastern cultural context.

**3. Image Quality (1–5).** (Same definition as in the Culture-Specific Experiment; see metric D.)

# E  DETAILED PROMPTS

The overall distribution of our prompt categories used in human evaluation is shown in Fig. 15. Our test dataset spans 15 semantic classes, covering a wide variety of concepts such as people, food, architecture, clothing, festivals, nature, mythology, and other culturally grounded or culturally neutral themes. This broad and balanced category distribution reflects that the prompts are not concentrated in a narrow region but instead provide extensive semantic coverage. Such diversity ensures that our

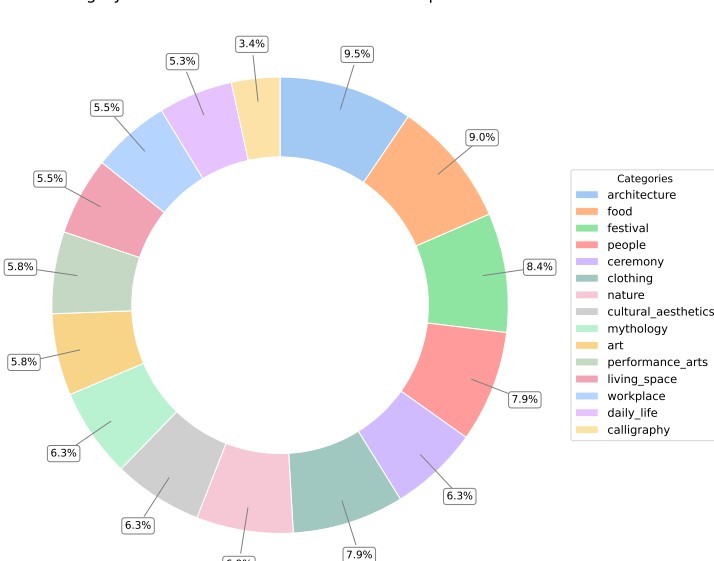

Figure 15: **Category distribution of our prompt set in human evaluation.** The 15 prompt categories cover a broad spectrum of concepts, including people, food, architecture, etc. The balanced distribution illustrates that our dataset spans diverse semantic regions rather than concentrating on a narrow subset, ensuring comprehensive coverage for both culture-specific and culture-neutral evaluations.

evaluation is comprehensive across both culture-specific and culture-neutral conditions, allowing models to be tested fairly on a wide range of visual and semantic scenarios.

As illustrated in the culture-specific word cloud (Fig. 16), the culture-specific prompts contain a rich collection of Chinese cultural concepts, including traditional rituals, historical architectural elements, classical activities, aesthetic descriptions, and symbolic natural imagery. These prompts capture nuanced cultural semantics and avoid relying on repetitive or templated phrasing, enabling a rigorous evaluation of a model's capability to understand and generate culturally grounded content.

In contrast, the culture-neutral word cloud (Fig. 17) highlights prompts that focus on everyday activities, general scenes, common objects, and universal human behaviors without embedding explicit cultural cues. Despite the absence of culture-dependent indicators, this subset still exhibits broad semantic diversity and substantial lexical variety, allowing us to analyze whether generative models unintentionally introduce cultural bias when the input itself is neutral.

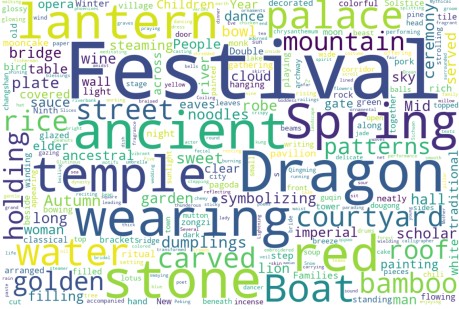

Figure 16: **Word cloud of culture-specific prompts.**

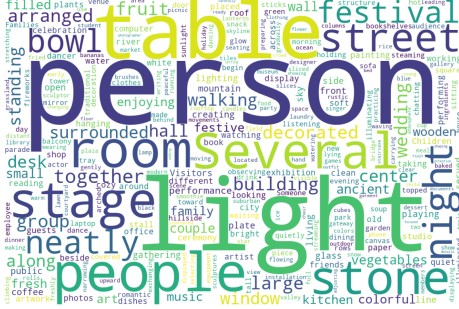

Figure 17: **Word cloud of culture-neutral prompts.**

---

**People**

EN: A person sitting at a desk, reading a book, with a warm lamp light illuminating the room
CN: 一个人坐在书桌前，阅读一本书，暖黄的灯光照亮了房间

EN: A person walking through a bustling city street, wearing a jacket and carrying a backpack
CN: 一个人走在繁忙的城市街道上，穿着夹克，背着背包

EN: A person jogging along a peaceful park path during the early morning hours
CN: 一个人清晨在宁静的公园小道上慢跑

EN: A person cooking in a modern kitchen, preparing a meal with fresh ingredients
CN: 一个人在现代化厨房里做饭，使用新鲜的食材

EN: A person enjoying a quiet moment at a coffee shop, sipping a cup of coffee while reading a newspaper
CN: 一个人在咖啡店里享受宁静的时光，喝着咖啡，阅读报纸

EN: A person standing in front of a large window, gazing outside at the rain
CN: 一个人站在大窗前，望着窗外的雨景

EN: A person painting on a canvas in a well-lit studio, surrounded by art supplies
CN: 一个人正在光线充足的工作室里画画，周围摆满了艺术用品

EN: A person biking on a trail through a forest, wearing a helmet and athletic gear
CN: 一个人骑车穿过森林的小径，戴着头盔，穿着运动装备

EN: A person shopping in a local market, selecting fresh fruits and vegetables
CN: 一个人在当地市场里选购新鲜的水果和蔬菜

EN: A person attending a conference, taking notes while listening to a speaker
CN: 一个人参加会议，边听演讲者发言边做笔记

EN: A person typing on a laptop in a cozy room with bookshelves in the background
CN: 一个人在舒适的房间里打字，背景是整齐的书架

EN: A person feeding a cat in a small kitchen
CN: 一个人在小厨房里喂猫

EN: A person lying on a sofa, listening to music with headphones
CN: 一个人躺在沙发上，戴着耳机听音乐

EN: A person walking a dog along a quiet suburban street
CN: 一个人在宁静的郊区街道上遛狗

EN: A person standing at a bus stop, holding an umbrella on a rainy day
CN: 一个人在下雨天站在公交车站，撑着伞

EN: A person browsing books in a public library
CN: 一个人在公共图书馆里浏览书籍

EN: A person assembling furniture in a new apartment
CN: 一个人在新公寓里组装家具

EN: A person looking at a phone while waiting in a line
CN: 一个人在排队等待时低头看手机

EN: A person climbing stairs in an office building
CN: 一个人在办公楼里爬楼梯

EN: A person brushing teeth in front of a mirror in the morning
CN: 一个人在早上对着镜子刷牙

EN: A person making a video call at a desk with a laptop
CN: 一个人坐在桌前用笔记本电脑进行视频通话

EN: A person tying shoelaces before going for a run
CN: 一个人在跑步前系鞋带

EN: A person folding laundry in a bright and clean room
CN: 一个人在明亮整洁的房间里叠衣服

EN: A person stretching on a yoga mat in a bright living room
CN: 一个人在明亮的客厅里垫子上做拉伸运动

EN: A person watering plants on a sunny balcony
CN: 一个人在阳光明媚的阳台上给植物浇水

Figure 18: Examples of text prompts from the *People* category used for testing cultural neutrality CLIP score. Each pair includes an English prompt and its corresponding Chinese translation. These prompts are designed to avoid cultural or regional bias while maintaining scene diversity and clarity.

Table 7: **Full list of bilingual prompts used in our culture-specific human experiments.**

| ID | English prompt | Chinese prompt |
|----|----------------|----------------|
| 0 | A woman wearing a traditional mamian skirt strolling through a classical courtyard. | 一位身穿马面裙的女子在古典庭院中漫步 |
| 1 | A scholar in a changshan reciting poems in a bamboo forest. | 一名身穿长衫的书生正在竹林里吟诗 |

| ID | English prompt | Chinese prompt |
|---|---|---|
| 2 | A Peking opera performer painting a facial mask backstage before going on stage. | 京剧演员在后台勾画脸谱，准备登台表演 |
| 3 | A farmer wearing a bamboo hat working in a rice field. | 一位农夫戴着斗笠在稻田里劳作 |
| 4 | A woman in a qipao walking along an old street while holding an umbrella. | 一位身穿旗袍的女子在老街上撑伞而行 |
| 5 | During a lion dance performance, two dancers are skillfully moving the colorful lion head. | 舞狮表演中，两名舞者正在挥动彩色的狮头 |
| 6 | A calligrapher writing with bold brush-strokes on a long scroll. | 书法家在长桌上挥毫泼墨 |
| 7 | A kung fu master practicing tai chi on the top of a mountain. | 功夫大师在山顶练习太极拳 |
| 8 | A man wearing a Tang-style jacket walking across a stone bridge in an ancient town. | 身穿唐装的男子正在古镇石桥上行走 |
| 9 | An elder in the village telling mythological stories to children. | 村庄里的老人正给孩子们讲述神话故事 |
| 10 | A fisherman casting a net by the riverbank. | 一名渔夫正在江边撒网捕鱼 |
| 11 | A monk meditating in front of a temple gate. | 一位僧人正坐在山门前打坐 |
| 12 | A musician playing the guqin in a courtyard. | 一位琴师在庭院中演奏古琴 |
| 13 | On the streets during Lunar New Year, children running happily with lanterns in hand. | 农历新年的街头，小孩提着灯笼欢快奔跑 |
| 14 | A magnificent palace standing on a raised platform, its roof covered with golden glazed tiles. | 一座宏伟的宫殿矗立在高台之上，屋顶覆盖着金色琉璃瓦 |
| 15 | An ancient city gate tower decorated with red lanterns. | 古老的城楼上悬挂着红色灯笼 |
| 16 | Artificial rock formations and flowing water complement each other in a classical garden. | 园林中的假山与流水相映成趣 |
| 17 | A stone bridge spanning a clear river, with patterns carved along its surface. | 石桥横跨在清澈的河流上，桥面雕刻着纹样 |
| 18 | On both sides of the old town's stone-paved street stand houses with white walls and dark tile roofs. | 古镇的青石板街道两侧是粉墙黛瓦的房屋 |
| 19 | An ornamental archway engraved with elaborate calligraphy and patterns. | 牌坊上刻有精美的书法和花纹 |
| 20 | Ancient city walls stretching across the landscape, thick and imposing. | 古代城墙绵延起伏，墙体厚实雄伟 |
| 21 | A winding corridor in a classical garden, decorated with vibrant paintings. | 园林里的长廊曲折蜿蜒，彩绘装饰鲜艳 |
| 22 | A Buddhist pagoda standing at the foot of a mountain, with each tier narrowing upward. | 佛塔耸立在山脚下，塔身逐层收窄 |
| 23 | An ancient opera stage with patterned curtains hanging under its eaves. | 古戏台的屋檐下悬挂着绘有图案的幕布 |
| 24 | An arched bridge in a water town reflecting in the river scenery. | 水乡小镇的拱桥与河道相映成景 |
| 25 | A palace's vermilion gate studded with golden door nails. | 宫殿的朱红色大门上钉满了金色门钉 |
| 26 | An ancient pagoda surrounded by stone lions and incense burners. | 古塔周围环绕着石狮和香炉 |
| 27 | The stone steps in front of an ancient temple worn smooth by time. | 古庙前的石阶被岁月磨得光滑 |
| 28 | In an imperial garden, the lake reflects pavilions and towers. | 皇家园林里湖面倒映着亭台楼阁 |

| ID | English prompt | Chinese prompt |
|----|----------------|----------------|
| 29 | The arches of an old bridge forming circular reflections on the water. | 古桥桥洞在水面上形成圆形倒影 |
| 30 | The beams and pillars of an ancestral hall carved with birds, dragons, and phoenixes. | 祠堂的梁柱上雕刻着花鸟与龙凤 |
| 31 | Mythical beasts lined up as decorations along the ridge of an ancient palace roof. | 古代宫殿的屋脊上排列着神兽装饰 |
| 32 | An ancient temple nestled among pine trees in the mountains. | 山间的古寺掩映在松树之间 |
| 33 | The palace roof ridge lined with mythical ridge beasts, symbolizing majesty and solemnity. | 宫殿屋脊上整齐排列着脊兽，彰显庄严气势 |
| 34 | Exquisite dougong brackets carved beneath the eaves of the main hall. | 大殿的屋檐下雕刻着精美的斗拱结构 |
| 35 | A decorated spirit wall painted with colorful patterns, concealing the inner courtyard view. | 影壁墙上绘有彩色图案，遮挡着院落的内部景致 |
| 36 | The beams of an ancient opera stage adorned with carved que-ti brackets. | 古戏楼的梁架上悬挂着雕刻的雀替装饰 |
| 37 | The roof of the imperial temple covered with green glazed tiles. | 皇家庙宇的屋顶覆盖着绿色琉璃瓦 |
| 38 | A stone paifang archway carved with dragons, phoenixes, and cloud motifs. | 石牌坊上雕刻着龙凤和祥云纹样 |
| 39 | A tall ornamental huabiao pillar in front of the temple, entwined with carved cloud dragons. | 寺庙门前的华表高耸，上面盘绕着雕刻的云龙 |
| 40 | Painted caisson patterns decorating the beams of a long corridor in a classical garden. | 园林长廊的梁枋上绘有彩色藻井图案 |
| 41 | Golden door studs arranged in nine rows and nine columns on the grand gate, symbolizing imperial dignity. | 大门的门钉呈九行九列排列，象征尊贵 |
| 42 | Chiwen ornaments on the ancestral hall roof gleaming under the sunlight. | 祠堂屋顶的鸱吻装饰在阳光下闪耀 |
| 43 | Tiered dougong brackets supporting the enormous eaves of the palace. | 宫殿的斗拱层层叠叠，支撑着巨大的屋檐 |
| 44 | Stone lions guarding both sides of the steps leading to the temple hall. | 寺庙大殿前的石狮守护在台阶两侧 |
| 45 | A winding covered bridge beside a waterside pavilion, its railings carved with floral patterns. | 园林水榭旁设有曲折的廊桥，栏杆雕刻着花草纹样 |
| 46 | A roof end beast on the palace gate appearing lifelike, mouth open as if roaring. | 宫门屋顶的吻兽张口欲啸，栩栩如生 |
| 47 | The top of the archway supported by dougong brackets holding a finely carved plaque. | 牌坊顶部的斗拱托举着雕刻精细的匾额 |
| 48 | A spirit wall in the palace courtyard inlaid with glazed bricks. | 宫殿中庭的影壁墙镶嵌着琉璃砖装饰 |
| 49 | Golden bells hanging beneath the eaves of an ancient pagoda, ringing in the breeze. | 古塔的屋檐下悬挂着金色铃铛，随风作响 |
| 50 | Coiling dragon motifs carved into the railings of the city tower. | 城楼的栏杆上雕刻着盘绕的蟠龙 |
| 51 | An octagonal pavilion at the end of the imperial garden corridor, its roof covered with glazed tiles. | 皇家园林的长廊尽头设有八角亭，亭顶覆盖着琉璃瓦 |

| ID | English prompt | Chinese prompt |
| --- | --- | --- |
| 52 | A steaming bowl of zhajiang noodles topped with rich soybean paste sauce and shredded cucumber. | 一碗热气腾腾的炸酱面，面条上覆盖着浓郁的酱料和黄瓜丝 |
| 53 | A Sichuan hotpot with boiling red chili oil, surrounded by plates of various ingredients. | 一桌四川火锅，红油翻滚，桌上摆满各种菜品 |
| 54 | A bamboo steamer filled with neatly arranged translucent shrimp dumplings. | 竹蒸笼里整齐摆放着晶莹剔透的虾饺 |
| 55 | A plate of kung pao chicken served with peanuts and dried chili peppers. | 一盘宫保鸡丁，配有花生和红辣椒 |
| 56 | A bowl of Lanzhou beef noodles with clear broth and chewy hand-pulled noodles. | 一碗兰州牛肉拉面，汤清味鲜，面条劲道 |
| 57 | Slices of crispy Beijing roast duck served with lotus leaf pancakes and sweet bean sauce. | 盘中摆放着酥脆的北京烤鸭片，旁边有荷叶饼和甜面酱 |
| 58 | A plate of mapo tofu in a bright red sauce, garnished with green onions. | 一盘麻婆豆腐，鲜红的汤汁里点缀着青葱 |
| 59 | A bamboo basket filled with steaming xiaolongbao dumplings, thin-skinned and juicy. | 竹篮里放着热腾腾的小笼包，皮薄汁多 |
| 60 | A steaming bowl of yangrou paomo with pieces of flatbread soaked in mutton soup. | 一碗热气腾腾的羊肉泡馍，配有白色馍块 |
| 61 | A plate of golden crispy spring rolls. | 盘中放着金黄酥脆的春卷 |
| 62 | A bowl of Chongqing noodles topped with chili oil and Sichuan peppercorns. | 一碗重庆小面，面上覆盖着辣椒油和花椒 |
| 63 | A plate of sweet and sour spare ribs with a glossy sauce. | 桌上摆放着香气扑鼻的糖醋排骨 |
| 64 | A plate of stir-fried water spinach, vibrant green and fresh. | 一盘清炒空心菜，翠绿鲜亮 |
| 65 | A bowl of rice porridge with preserved egg and minced pork. | 一碗粥里放着皮蛋和瘦肉 |
| 66 | A dish of yuxiang shredded pork with a glossy red sauce. | 桌上摆放着一份鱼香肉丝，色泽红润 |
| 67 | A plate of handmade dumplings neatly arranged. | 碟子里盛放着手工制作的饺子，排列整齐 |
| 68 | A bamboo steamer containing shumai dumplings topped with small pieces of carrot. | 竹蒸笼里有几只烧卖，表面点缀着胡萝卜粒 |
| 69 | A bowl of Guilin rice noodles served with pickled long beans and crushed peanuts. | 一碗桂林米粉，配有酸豆角和花生碎 |
| 70 | A large dish of braised chicken with potatoes and wide noodles, rich in flavor. | 大盘鸡里有土豆块和宽面条，鲜香浓郁 |
| 71 | A plate of braised pork belly in soy sauce, tender and glossy red. | 盘中是一份红烧肉，肥而不腻，色泽红亮 |
| 72 | Steaming dumplings served on the Spring Festival table, symbolizing family reunion. | 春节餐桌上的饺子，热气腾腾，象征团圆 |
| 73 | Glutinous rice balls boiling in hot water during the Lantern Festival, smooth and round. | 元宵节的汤圆在热水里翻滚，表面光滑圆润 |
| 74 | Zongzi wrapped in bamboo leaves for the Dragon Boat Festival, neatly shaped. | 端午节的粽子被用粽叶包裹，形状整齐 |
| 75 | Mooncakes cut into small pieces during the Mid-Autumn Festival, with sweet and rich fillings. | 中秋节的月饼切成小块，馅料香甜可口 |
| 76 | Chrysanthemum wine for the Double Ninth Festival, giving off a delicate fragrance. | 重阳节的菊花酒散发出淡雅清香 |
| 77 | Dumplings neatly arranged on the board, ready to be boiled for the Winter Solstice. | 冬至的饺子整齐地摆放在案板上，等待下锅 |

| ID | English prompt | Chinese prompt |
|---|---|---|
| 78 | Fish served at the New Year's Eve dinner, symbolizing abundance every year. | 春节年夜饭上的鱼，寓意年年有余 |
| 79 | Five-kernel mooncake for Mid-Autumn Festival, revealing a rich nut filling when cut open. | 中秋节的五仁月饼切开后露出丰富的果仁馅 |
| 80 | Savory pork zongzi for Dragon Boat Festival, fragrant with bamboo leaves and sticky rice. | 端午节的咸肉粽散发着粽叶与糯米的香味 |
| 81 | People enjoying sweet rice balls beside lanterns during the Lantern Festival. | 元宵节的花灯旁，人们分享甜汤圆 |
| 82 | Slices of sticky rice cake served on the Spring Festival table, soft and sweet. | 春节餐桌上的年糕被切成片，软糯香甜 |
| 83 | A steaming bowl of mutton soup for the Winter Solstice, warming and nourishing. | 冬至的一碗羊肉汤热气腾腾，暖胃驱寒 |
| 84 | Snow skin mooncakes during Mid-Autumn Festival, pastel-colored and chewy. | 中秋节的冰皮月饼颜色清新，口感软糯 |
| 85 | Candied hawthorn skewers served during Spring Festival, crystal clear and sweet-sour. | 春节餐桌上的糖葫芦晶莹剔透，酸甜可口 |
| 86 | Alkaline water zongzi with a pale yellow color, eaten during Dragon Boat Festival. | 端午节的碱水粽呈现淡黄色，口感清香 |
| 87 | Layered cakes for the Double Ninth Festival, symbolizing rising step by step. | 重阳节的糕点层层叠叠，寓意步步高升 |
| 88 | Sesame-filled rice balls for the Lantern Festival, oozing rich filling when cut open. | 元宵节的芝麻馅汤圆切开后流出香浓馅料 |
| 89 | Longevity noodles on the Spring Festival table, symbolizing health and long life. | 春节餐桌上的长寿面寓意健康长久 |
| 90 | Lotus seed paste mooncake for Mid-Autumn Festival, golden crust with delicate filling. | 中秋节的莲蓉月饼外皮金黄，内馅细腻 |
| 91 | Sweet red date zongzi for the Dragon Boat Festival, fragrant and pleasantly sweet. | 端午节的红枣粽香气扑鼻，甜而不腻 |
| 92 | Streets decorated with red lanterns and couplets during the Spring Festival. | 春节的街道上挂满红灯笼和春联 |
| 93 | People setting off fireworks and firecrackers on New Year's Eve. | 人们在除夕夜燃放烟花爆竹 |
| 94 | Colorful lanterns hanging high in the square during the Lantern Festival night. | 元宵节的夜晚，广场上高挂彩色花灯 |
| 95 | During the Qingming Festival, people offering food and incense at ancestors' graves. | 清明时节，人们在墓前祭祖，摆放供品 |
| 96 | Dragon boats racing on the river during the Dragon Boat Festival, with thunderous drumbeats. | 端午节龙舟在江面上竞速，鼓声震天 |
| 97 | Families gathering in the courtyard to admire the full moon and eat mooncakes during the Mid-Autumn Festival. | 中秋节的夜晚，人们在庭院里赏月吃月饼 |
| 98 | Families making dumplings in the kitchen during the Winter Solstice. | 冬至时节，家家户户在厨房里包饺子 |
| 99 | Crowds filling a Spring Festival temple fair, with vendors selling sugar figurines. | 春节庙会上人群熙攘，摊位上售卖糖人 |
| 100 | Lion dance performances attracting crowds on the streets during the Lantern Festival. | 元宵节的街头有舞狮表演，吸引观众围观 |
| 101 | Families hanging mugwort and calamus on their doors during the Dragon Boat Festival. | 端午节家门口悬挂艾草和菖蒲 |
| 102 | Children carrying colorful lanterns while strolling on the streets during Mid-Autumn night. | 中秋夜小孩手提彩灯在街头游玩 |

| ID | English prompt | Chinese prompt |
|---|---|---|
| 103 | Dragon dance teams performing in the square during the Spring Festival. | 春节舞龙队伍在广场上盘旋起舞 |
| 104 | Qingtuan glutinous rice dumplings with bright green skins sold in markets during Qingming Festival. | 清明节市场上售卖青团，绿油油的外皮十分鲜亮 |
| 105 | Families enjoying chrysanthemum wine and Double Ninth cakes during the festival. | 重阳节人们在家里品尝菊花酒和重阳糕 |
| 106 | Children helping elders put New Year paintings on the wall during the Spring Festival. | 春节贴年画的场景，小孩帮着长辈贴在墙上 |
| 107 | A Mid-Autumn Festival table with fruits and osmanthus wine prepared for moon gazing. | 中秋节摆放着水果和桂花酒的赏月桌 |
| 108 | Lion dance troupes parading through the streets with loud drums and gongs during Spring Festival. | 春节的舞狮队伍走街串巷，锣鼓喧天 |
| 109 | A steaming pot of mutton soup on the dinner table during the Winter Solstice night. | 冬至夜晚的餐桌上热气腾腾的羊肉汤飘香 |
| 110 | A woman wearing a qipao embroidered with flowers and birds, standing on an old street. | 一名女子身穿绣有花鸟图案的旗袍，站在古街上 |
| 111 | A man in a changshan holding a folding fan, strolling in a courtyard. | 一名男子身着长衫，手持折扇在庭院中漫步 |
| 112 | A dancer wearing traditional Miao attire, with silver ornaments shining under the lights. | 舞蹈演员身着苗族盛装，银饰在灯光下闪耀 |
| 113 | An actor in Peking opera costume with painted facial makeup. | 演员穿着京剧戏服，脸上勾画着脸谱 |
| 114 | A woman wearing a Su embroidery shawl with patterns as delicate as paintings. | 女子穿着苏绣披肩，花纹细腻如画 |
| 115 | A man wearing a black long robe with a jade pendant at his waist. | 男子身着黑色长袍，腰系玉佩 |
| 116 | A bride wearing a red xiuhe dress with a traditional bridal veil. | 一位新娘身穿红色秀禾服，头戴喜帕 |
| 117 | A woman in a Han dynasty quju robe with flowing hems. | 女子穿着汉代曲裾深衣，衣摆飘逸 |
| 118 | A man wearing a Ming dynasty round-collar robe with cloud patterns on the shoulders. | 男子穿着明代圆领袍，肩部装饰着云纹 |
| 119 | A dancer in a Uyghur traditional dress with a skirt swirling during the dance. | 舞者穿着维吾尔族长裙，裙摆在舞动中旋转 |
| 120 | A Mongolian man wearing a long robe with a wide leather belt around his waist. | 蒙古族男子穿着长袍，腰间系着宽皮带 |
| 121 | Children wearing traditional tiger-head hats and dudou playing in the courtyard. | 儿童穿着传统虎头帽和肚兜，在院子里玩耍 |
| 122 | An elder in a cloth changshan resting at the village entrance with a cane. | 一位老者穿着布衣长衫，拄着拐杖在村口休息 |
| 123 | An actor in Kunqu opera costume with wide, flowing sleeves. | 演员穿着昆曲华服，衣袖宽大飘逸 |
| 124 | A woman in a brocade robe woven with gold threads, embroidered with peony patterns. | 女子穿着织金锦袍，绣有牡丹花纹 |
| 125 | A bride and groom performing the bowing ceremony at a wedding, with red candles burning brightly. | 新郎新娘在婚礼上行拜堂礼，红烛高燃 |
| 126 | Elders seated in the hall while the newlyweds serve tea to their parents. | 长辈端坐在高堂，新人向父母敬茶 |
| 127 | People burning incense and praying for blessings during a temple fair ritual. | 庙会祭祀中，人们焚香祈福，香烟缭绕 |

| ID | English prompt | Chinese prompt |
|---|---|---|
| 128 | Families presenting offerings at ancestral graves during the Qingming Festival. | 清明祭祖时，家人在墓前献上供品 |
| 129 | A Taoist priest in ceremonial robes conducting a ritual before the altar. | 道士身着法衣在坛前做法事 |
| 130 | Dragon dance teams performing during a festive ceremony, accompanied by thunderous drums. | 舞龙队伍在节日仪式中起舞，鼓声雷动 |
| 131 | During the coming-of-age ceremony in the village, a young man wears a symbolic headpiece. | 村里的成年礼上，少年戴上象征成年的头冠 |
| 132 | Temple bells tolling as monks chant scriptures together. | 寺庙钟声敲响，僧侣齐声诵经 |
| 133 | Examinees bowing to the chief examiner during the imperial examination palace test. | 考生在科举殿试中向主考官行礼 |
| 134 | A bride wearing a red bridal veil shyly lowering her head during the bowing ceremony. | 新娘头戴红色喜帕，在拜堂仪式上羞涩低头 |
| 135 | Villagers chanting sacrificial texts together during a ritual at the village temple. | 村庙前的祭祀仪式上，乡民齐声诵读祭文 |
| 136 | A grand investiture ceremony taking place in the imperial palace hall, accompanied by ritual music. | 皇宫大殿内举行盛大的册封礼，礼乐齐鸣 |
| 137 | An eternal lamp lit before ancestral tablets, with family members solemnly paying respects. | 祖先牌位前点燃长明灯，家族成员肃然敬拜 |
| 138 | A groom stepping over a brazier during the wedding ceremony, symbolizing the warding off of evil. | 新郎在婚礼上跨过火盆，寓意驱邪避灾 |
| 139 | Elders holding chrysanthemum wine during the Double Ninth Festival climbing ceremony, praying for blessings. | 重阳节登高仪式上，长者手持菊花酒祈福 |
| 140 | Monks sprinkling holy water with willow branches during a temple consecration ritual. | 庙宇的开光仪式中，僧人手持柳枝洒净 |
| 141 | Young scholars parading on horseback through the streets during the top-scorer celebration. | 状元及第的庆典上，年轻学子骑马游街 |
| 142 | A winter ancestral worship ceremony in the clan's ancestral hall, with descendants bowing in order. | 家族祠堂里举行冬祭，子孙依次叩拜 |
| 143 | During the Dragon Boat Festival river sacrifice, people casting offerings into the water. | 端午节祭江仪式上，人们向江水抛洒供品 |
| 144 | Nüwa mending the sky with five-colored stones as the cracks gradually close. | 女娲手持五色石补天，天空裂缝渐渐合拢 |
| 145 | Pangu splitting chaos with a giant axe to create heaven and earth. | 盘古开天辟地，手持巨斧劈开混沌 |
| 146 | Chang'e ascending to the Moon Palace in white robes, accompanied by a jade rabbit. | 嫦娥身着白衣飞升月宫，身旁伴有玉兔 |
| 147 | Hou Yi drawing his bow to shoot down the blazing suns in the sky. | 后羿拉弓射日，天空中炽热的太阳逐个坠落 |
| 148 | The Cowherd and Weaver Girl gazing across the Milky Way, with a magpie bridge connecting them. | 牛郎与织女隔着银河相望，鹊桥横跨其间 |
| 149 | Yu the Great controlling floods along the rivers. | 大禹正在江河边疏导洪水 |
| 150 | Fuxi holding a bagua diagram, with dragons and serpents coiling beside him. | 伏羲手持八卦图，身旁盘绕着龙蛇 |

| ID | English prompt | Chinese prompt |
|---|---|---|
| 151 | Nezha riding on wind-fire wheels while wielding the cosmic scarf. | 哪吒脚踏风火轮，手持混天绫 |
| 152 | Guanyin Bodhisattva holding a vase and willow branch, standing on a lotus pedestal. | 观音菩萨手持净瓶与杨柳枝，立于莲花座上 |
| 153 | Erlang Shen with a shining third eye on his forehead, holding a three-pointed double-edged blade. | 二郎神额头竖眼闪烁光芒，手持三尖两刃刀 |
| 154 | The Eight Immortals crossing the sea, each displaying their magical powers amidst surging waves. | 八仙渡海，各显神通，波涛汹涌 |
| 155 | The Yellow Emperor confronting Chiyou in the fierce Battle of Zhuolu. | 黄帝与蚩尤在涿鹿之战中激烈对峙 |
| 156 | Jingwei transformed into a bird, tirelessly carrying twigs and stones to fill the sea. | 精卫化作小鸟，不断衔木石填海 |
| 157 | Kua Fu chasing the sun, running across the land with a wooden staff. | 夸父追逐太阳，手持木杖奔跑在大地上 |
| 158 | Bai Suzhen from the Legend of the White Snake, transformed into human form holding a paper umbrella. | 白蛇传中的白素贞化身人形，手持纸伞 |
| 159 | Fairy maidens dancing in the Jade Pool while the Queen Mother of the West sits in the center. | 精灵般的仙女在瑶池起舞，王母娘娘端坐其中 |
| 160 | The Earth Goddess Houtu stepping on the ground, lifting mountains and rivers with her hands. | 后土女神脚踏大地，双手托举山河 |
| 161 | The Thunder God and Lightning Goddess wielding drums and lightning amidst dark clouds. | 雷公电母在乌云中挥舞鼓槌和闪电 |
| 162 | Chenxiang splitting the mountain to rescue his mother, with rocks scattering in all directions. | 沉香劈山救母，山石飞溅四散 |
| 163 | A calm lake reflecting the blue sky and white clouds, with gentle ripples caused by a light breeze. | 宁静的湖面倒映着蓝天和白云，微风吹起涟漪 |
| 164 | Wildflowers blooming on a plateau, swaying in the wind against the backdrop of a vast grassland. | 高原上盛开的野花在风中摇曳，背景是一片广阔的草地 |
| 165 | Ocean waves crashing against coastal rocks, with splashes of water sparkling in the sunlight. | 海岸边的岩石被海浪拍打，水花在阳光下闪耀 |
| 166 | A star-filled night sky, with distant snow-covered mountains clearly visible under the moonlight. | 夜空中点缀着繁星，远处的雪山在月光下显得格外清晰 |
| 167 | A forest in autumn with leaves turning shades of red and yellow. | 秋天的森林里，树叶呈现出红黄相间的色彩 |
| 168 | Rolling sand dunes in the desert glowing golden under the setting sun. | 沙漠里起伏的沙丘在夕阳下泛着金光 |
| 169 | A glacier crevasse glowing with a faint blue light. | 冰川裂缝中透出幽蓝的光芒 |
| 170 | A waterfall cascading down from a height in the tropical rainforest. | 热带雨林里瀑布从高处倾泻而下 |
| 171 | Flocks of wild birds circling above the plains. | 平原上成群的野鸟在天空中盘旋 |
| 172 | Thick white smoke rising from a volcanic crater. | 火山口冒出滚滚白烟 |

| ID | English prompt | Chinese prompt |
|---|---|---|
| 173 | A clear stream winding through a mountain valley. | 清澈的溪流在山谷间蜿蜒流淌 |
| 174 | The northern lights dancing across the Arctic night sky. | 北极的极光在夜空中舞动 |
| 175 | Dark storm clouds gathering in the sky before a thunderstorm. | 暴风雨前，天空中乌云密布 |
| 176 | The ocean horizon glowing with golden light at dawn. | 晨曦下的海平面泛起一层金色光辉 |
| 177 | Steep and rugged cliffs rising on both sides of a canyon. | 峡谷两侧的岩壁陡峭险峻 |
| 178 | A rainbow appearing over the mountains after the rain. | 雨后，山间出现一道彩虹 |
| 179 | A frozen lake shimmering under the sunlight. | 冰封的湖面在阳光下闪闪发光 |
| 180 | Scholars sitting together brewing tea, with bamboo shadows swaying outside the window. | 文人雅士对坐煮茶，窗外竹影摇曳 |
| 181 | A calligrapher writing boldly on a long scroll, with the fragrance of ink filling the air. | 书法家在长桌上挥毫泼墨，墨香四溢 |
| 182 | A painter working in the courtyard with paper spread out, while birds sing in the distance. | 画师在庭院中铺纸作画，远处传来鸟鸣 |
| 183 | A guqin resting on a stone table, with a clear spring flowing nearby. | 古琴横放在石桌上，清泉在旁潺潺流淌 |
| 184 | A poet gazing afar from a tower, reciting verses with wine in hand. | 诗人登楼远眺，把酒临风吟诵诗句 |
| 185 | A winding garden path leading to a secluded stone bench and pavilion. | 园林的曲径通幽处设有石凳与小亭 |
| 186 | A scholar playing the guqin beneath pine trees, harmonizing with the sound of the wind. | 文士在松树下抚琴，松涛与琴声相和 |
| 187 | A lady in the garden picking flowers and placing petals into a sachet. | 仕女在花园中折花，把花瓣放入香囊 |
| 188 | Several students playing a game of Go deep within a bamboo grove. | 竹林深处几位学子对弈围棋 |
| 189 | A small boat drifting under the moon, with candlelight flickering inside. | 月下小舟漂泊，船中烛光摇曳 |
| 190 | Scholars gathering, clinking wine cups, and conversing about landscapes and life. | 文人聚会，酒杯相碰，畅谈山水与人生 |
| 191 | A scholar reading ancient texts inside a pavilion near a stone bridge. | 石桥旁的亭子里，学者正低头研读古籍 |
| 192 | A courtyard adorned with bonsai and Taihu rocks, exuding a tranquil charm. | 庭院中摆放着盆景和太湖石，气韵悠然 |
| 193 | A refined gathering where several people sit together composing poems and painting. | 雅集场景，几人围坐吟诗作画 |
| 194 | A lady setting paper boats afloat by the pond under a gentle breeze. | 清风吹拂下，仕女在池边放纸船 |
| 195 | A recluse leaning against a pine tree, appearing calm and carefree. | 高士倚松而立，神情悠然自得 |
| 196 | A poet writing verses during the blossoming of plum blossoms. | 梅花盛开之际，诗人提笔留下题咏 |
| 197 | A monk meditating on the stone steps beside an ancient pagoda. | 古塔旁的石阶上，一位僧人正在打坐 |
| 198 | A scholar sipping wine while watching the rain, as water drips from the eaves. | 文人把盏观雨，檐下水珠不断滴落 |

| ID | English prompt | Chinese prompt |
|---|---|---|
| 199 | A fisherman pulling in his net by the river-bank, while the sound of a shepherd's flute drifts from afar. | 渔夫在江边收网，远处传来牧笛声 |

Table 8: **Full list of bilingual prompts used in our culture-neutral human experiments.**

| ID | English prompt | Chinese prompt |
|---|---|---|
| 0 | An artist is working on a canvas, adding details with focused attention. | 一名艺术家正在画布上创作 |
| 1 | A gallery space with several paintings on display, arranged neatly along the walls. | 画廊中摆放着几幅正在展出的绘画作品 |
| 2 | A sculptor in a studio carefully shaping a stone piece with tools. | 一名雕塑家在工作室里打磨石头作品 |
| 3 | A large art installation placed at the center of an exhibition hall, surrounded by visitors observing it. | 展览厅中央放置着一件大型艺术装置 |
| 4 | A photographer setting up lighting equipment, preparing to capture a subject. | 一位摄影师正在布置灯光准备拍摄 |
| 5 | A student practicing drawing in an art classroom, with sketchbooks and pencils on the desk. | 一名学生在美术教室里学习绘画 |
| 6 | Visitors in the main hall of an art museum, observing the artworks displayed around them. | 艺术博物馆的大厅里，观众正在欣赏展出的作品 |
| 7 | An artist mixing different colors on a palette at a worktable. | 一名艺术家在工作台上调配颜色 |
| 8 | A designer creating a digital artwork on a computer screen, surrounded by graphic tools. | 一位设计师在电脑上创作数字艺术作品 |
| 9 | An open exhibition hall displaying artworks made from various materials, placed across the space. | 开放式展厅里摆放着不同材质的艺术作品 |
| 10 | A person sitting at a desk, reading a book, with a warm lamp light illuminating the room | 一个人坐在书桌前，阅读一本书，暖黄的灯光照亮了房间 |
| 11 | A person walking through a bustling city street, wearing a jacket and carrying a backpack | 一个人走在繁忙的城市街道上，穿着夹克，背着背包 |
| 12 | A person jogging along a peaceful park path during the early morning hours | 一个人清晨在宁静的公园小道上慢跑 |
| 13 | A person cooking in a modern kitchen, preparing a meal with fresh ingredients | 一个人在现代化厨房里做饭，使用新鲜的食材 |
| 14 | A person enjoying a quiet moment at a coffee shop, sipping a cup of coffee while reading a newspaper | 一个人在咖啡店里享受宁静的时光，喝着咖啡，阅读报纸 |
| 15 | A person standing in front of a large window, gazing outside at the rain | 一个人站在大窗前，望着窗外的雨景 |
| 16 | A person painting on a canvas in a well-lit studio, surrounded by art supplies | 一个人正在光线充足的工作室里画画，周围摆满了艺术用品 |
| 17 | A person biking on a trail through a forest, wearing a helmet and athletic gear | 一个人骑车穿过森林的小径，戴着头盔，穿着运动装备 |
| 18 | A person shopping in a local market, selecting fresh fruits and vegetables | 一个人在当地市场里选购新鲜的水果和蔬菜 |

| ID | English prompt | Chinese prompt |
|---|---|---|
| 19 | A person attending a conference, taking notes while listening to a speaker | 一个人参加会议，边听演讲者发言边做笔记 |
| 20 | A person typing on a laptop in a cozy room with bookshelves in the background | 一个人在舒适的房间里打字，背景是整齐的书架 |
| 21 | A person stretching on a yoga mat in a bright living room | 一个人在明亮的客厅里垫子上做拉伸运动 |
| 22 | A person watering plants on a sunny balcony | 一个人在阳光明媚的阳台上给植物浇水 |
| 23 | A person feeding a cat in a small kitchen | 一个人在小厨房里喂猫 |
| 24 | A person lying on a sofa, listening to music with headphones | 一个人躺在沙发上，戴着耳机听音乐 |
| 25 | A person walking a dog along a quiet suburban street | 一个人在宁静的郊区街道上遛狗 |
| 26 | A person standing at a bus stop, holding an umbrella on a rainy day | 一个人在下雨天站在公交车站，撑着伞 |
| 27 | A person browsing books in a public library | 一个人在公共图书馆里浏览书籍 |
| 28 | A person assembling furniture in a new apartment | 一个人在新公寓里组装家具 |
| 29 | A person looking at a phone while waiting in a line | 一个人在排队等待时低头看手机 |
| 30 | A person climbing stairs in an office building | 一个人在办公楼里爬楼梯 |
| 31 | A person brushing teeth in front of a mirror in the morning | 一个人在早上对着镜子刷牙 |
| 32 | A person making a video call at a desk with a laptop | 一个人坐在桌前用笔记本电脑进行视频通话 |
| 33 | A person tying shoelaces before going for a run | 一个人在跑步前系鞋带 |
| 34 | A person folding laundry in a bright and clean room | 一个人在明亮整洁的房间里叠衣服 |
| 35 | A modern skyscraper with reflective glass windows standing tall against a clear blue sky | 一座现代化的摩天大楼，玻璃窗反射着清澈的蓝天 |
| 36 | A cozy cottage surrounded by a lush garden, with a small porch and a white picket fence | 一座温馨的小屋，四周是郁郁葱葱的花园，门前有一个小露台和白色栅栏 |
| 37 | A quaint coffee shop with vintage décor, wooden tables, and cozy seating arrangements | 一个别致的咖啡店，复古风格的装饰，木制桌子和舒适的座位 |
| 38 | An old building with ivy climbing up its walls, located in a historic district | 一座古老的建筑，爬满了常春藤，位于历史街区 |
| 39 | A minimalist home with clean lines, large windows, and an open-concept living area | 一座极简主义风格的家，简洁的线条，大窗户和开放式客厅 |
| 40 | A public library with rows of bookshelves, reading tables, and large windows letting in natural light | 一个公共图书馆，书架整齐排列，设有阅读桌，窗外自然光洒入 |
| 41 | A classical building with ornate columns, arches, and sculptures | 一座典雅的古典风格建筑，装饰精美的柱子、拱门和雕塑 |
| 42 | A suburban house with a neatly trimmed lawn, two-car garage, and a red front door | 一栋郊区住宅，草坪修剪整齐，有双车车库和红色前门 |
| 43 | A beach house on wooden stilts, facing the ocean, with a wide deck and hammock | 一座建在木桩上的海边别墅，面朝大海，有宽敞的平台和吊床 |
| 44 | A rustic barn with wooden beams and a large sliding door, surrounded by open fields | 一座乡村谷仓，有粗大的木梁和大滑门，四周是一望无际的田野 |
| 45 | A mountain lodge made of stone and timber, with smoke rising from the chimney | 一间由石材和木材建成的山间小屋，烟囱里升起袅袅炊烟 |
| 46 | A rooftop garden on a city building, with plants, benches, and a view of the skyline | 一座城市建筑的屋顶花园，有植物、长椅和城市天际线的美景 |

| ID | English prompt | Chinese prompt |
|----|----------------|----------------|
| 47 | A university campus with brick buildings, clock tower, and green lawns | 一座大学校园，有红砖建筑、钟楼和绿色草坪 |
| 48 | A lakeside villa with large glass doors opening to a wooden dock | 一栋湖边别墅，有大玻璃门通向木制码头 |
| 49 | A cozy mountain cabin with a slanted roof, warm lights, and snow on the ground | 一间温馨的山间木屋，屋顶倾斜，灯光暖黄，地上覆盖着积雪 |
| 50 | An indoor sports arena with a curved roof, stadium seating, and a basketball court | 一座室内体育馆，有弯曲的屋顶、阶梯看台和篮球场 |
| 51 | An old courtyard surrounded by tall walls, with an open central yard. | 一座古老的庭院被高墙围绕，中间有一个开阔的院落 |
| 52 | A stone bridge stretches across the river, with water flowing gently beneath it. | 石桥横跨在河面上，桥下的水缓缓流动 |
| 53 | A traditional wooden structure with a roof covered in neatly arranged tiles. | 传统木结构建筑，屋顶覆盖着整齐的瓦片 |
| 54 | An ancient tower standing on a hillside, with layered structures gradually narrowing upward. | 一座古老的塔楼屹立在山坡上，层层结构逐渐收窄 |
| 55 | An arched gateway in the center of a plaza, leading toward a cluster of buildings behind it. | 广场中央有一座拱形大门，通往后方的建筑群 |
| 56 | A hot bowl of soup with chunks of vegetables, meat, and noodles, served in a rustic ceramic bowl | 一碗热气腾腾的汤，里面有蔬菜块、肉块和面条，盛在古朴的陶瓷碗中 |
| 57 | A basket of freshly baked bread rolls with a side of butter and jam | 一篮新鲜出炉的面包卷，配有黄油和果酱 |
| 58 | A sushi platter with assorted pieces of nigiri, sashimi, and rolls, served with soy sauce and wasabi | 一盘寿司，包含各种生鱼片、寿司卷和手握寿司，配有酱油和芥末 |
| 59 | A steaming bowl of rice topped with stir-fried vegetables and a fried egg | 一碗热腾腾的米饭，上面放着炒蔬菜和煎蛋 |
| 60 | A smoothie bowl topped with granola, fresh fruits, and a drizzle of honey | 一碗冰沙，顶部撒上格兰诺拉麦片、新鲜水果和一丝蜂蜜 |
| 61 | A fresh garden salad with leafy greens, tomatoes, and a light dressing | 一份新鲜的蔬菜沙拉，包含生菜、番茄和清淡调味汁 |
| 62 | A bowl of hot soup with mixed vegetables and herbs, steaming gently | 一碗热腾腾的汤，混合蔬菜和香草，轻轻冒着热气 |
| 63 | A plate of roasted vegetables with crispy edges on a wooden table | 一盘边缘微焦的烤蔬菜，摆在木质桌面上 |
| 64 | A stack of fluffy pancakes topped with syrup and fruit slices | 一叠松软的煎饼，上面淋着糖浆和水果片 |
| 65 | A glass of fruit smoothie served with a straw and ice cubes | 一杯果味冰沙，配有吸管和冰块 |
| 66 | A baked pastry on a white plate, with golden crust and soft filling | 一块烘焙点心，金黄的外壳包裹着柔软的内馅，放在白色盘子上 |
| 67 | A cup of hot beverage beside a plate of small biscuits | 一杯热饮放在装有小饼干的盘子旁 |
| 68 | A fruit bowl with apples, bananas, and grapes on a kitchen counter | 厨房台面上的一个水果碗，里面有苹果、香蕉和葡萄 |
| 69 | A bowl of rice served with a side of cooked vegetables | 一碗米饭配上一份熟蔬菜 |
| 70 | A dessert plate with a small cake and a dollop of whipped cream | 甜点盘上有一块小蛋糕和一团鲜奶油 |
| 71 | A clear glass of cold water with lemon slices | 一杯透明的柠檬水，杯中漂浮着几片柠檬 |

| ID | English prompt | Chinese prompt |
| --- | --- | --- |
| 72 | A snack tray with nuts, dried fruits, and crackers | 一盘小吃，包含坚果、干果和薄脆饼干 |
| 73 | A plain bowl of cereal with milk and sliced bananas | 一碗简单的麦片，加了牛奶和香蕉片 |
| 74 | A parade with a giant dragon and dancers moving through a street decorated with banners and colorful lights | 游行队伍中，巨龙与舞者穿过布满横幅和彩灯的街道 |
| 75 | A family gathering around a dinner table during a traditional festival, with dishes shared among the group | 传统节日中，一家人围坐在餐桌旁，共享丰盛的佳肴 |
| 76 | A romantic evening with a couple under a full moon, enjoying a picnic and exchanging gifts | 一对情侣在明亮的圆月下享受野餐，互赠礼物，浪漫至极 |
| 77 | Children running around with lanterns during a night festival, laughing and playing together | 夜晚的节日中，孩子们手持灯笼，笑着玩耍 |
| 78 | Families flying colorful kites on a windy day during a cultural holiday | 文化节日里，家庭成员在风和日丽的日子里放飞色彩缤纷的风筝 |
| 79 | A boat race with decorated vessels on a river during a seasonal festival | 节庆期间，装饰华丽的船只在河上竞渡 |
| 80 | Elders gathering in the mountains, enjoying tea and nature on a festive day | 节日里，老人们在山间聚集，品茶享受自然风光 |
| 81 | A street lined with decorations for a holiday celebration, with families and friends walking together | 节日庆典时，街道两旁装饰着彩带，家庭和朋友们并肩走过 |
| 82 | A night festival with lanterns floating on a river, illuminating the water with soft light | 夜晚的节日，灯笼漂浮在河面上，温柔的光辉照亮水面 |
| 83 | People enjoying a public celebration, watching fireworks and eating traditional snacks | 人们在公开庆典中，观看烟花表演，享受传统小吃 |
| 84 | People gathering in a city square for a public celebration with lights and music | 人们聚集在城市广场上，灯光与音乐营造出热闹的庆典氛围 |
| 85 | A group of friends watching fireworks together under a night sky | 一群朋友在夜空下仰望烟花绽放 |
| 86 | Families enjoying a picnic in the park with balloons and festive decorations | 家庭成员在公园里野餐，四周布置着气球和节日装饰 |
| 87 | Children running through a field during a festive outdoor event | 节日活动中，孩子们在草地上欢快奔跑 |
| 88 | A quiet street illuminated with string lights and festive colors | 安静的街道上挂满了彩灯和节日色彩装饰 |
| 89 | People dancing in a parade with colorful costumes and cheerful music | 游行中人们穿着鲜艳服饰伴随欢乐音乐跳舞 |
| 90 | A decorated table with candles, desserts, and drinks during a celebration | 庆典中摆满甜点、饮品与蜡烛的装饰餐桌 |
| 91 | A family taking photos together in front of a decorated house | 一家人在装饰过的房屋前合影留念 |
| 92 | People walking along a waterfront at night during a festival of lights | 节日灯光秀中，人们夜晚漫步在水边 |
| 93 | Couples enjoying a romantic dinner with dim lighting and soft music | 情侣在昏黄灯光和柔和音乐中共进浪漫晚餐 |
| 94 | Kids playing with glow sticks and bubbles at an evening gathering | 孩子们在夜晚聚会中玩耍荧光棒和泡泡 |
| 95 | People sitting on a hillside watching a light show in the distance | 人们坐在山坡上远望灯光秀 |
| 96 | A neighborhood organizing a street fair with food stalls and games | 街区举行嘉年华，设有美食摊位与游乐项目 |

| ID | English prompt | Chinese prompt |
|---|---|---|
| 97 | A rooftop party with hanging lights and people chatting under the stars | 屋顶派对上悬挂着灯串，人们在星空下交谈 |
| 98 | A group of people lighting candles together in a peaceful evening ceremony | 一群人在宁静的傍晚共同点亮蜡烛，气氛祥和 |
| 99 | A misty valley in the early morning, with sunlight streaming down from distant mountain peaks. | 清晨的山谷中弥漫着薄雾，阳光从远处的山峰洒落下来 |
| 100 | A calm lake reflecting the blue sky and white clouds, with gentle ripples caused by a light breeze. | 宁静的湖面倒映着蓝天和白云，微风吹起涟漪 |
| 101 | Wildflowers blooming on a plateau, swaying in the wind against the backdrop of a vast grassland. | 高原上盛开的野花在风中摇曳，背景是一片广阔的草地 |
| 102 | Ocean waves crashing against coastal rocks, with splashes of water sparkling in the sunlight. | 海岸边的岩石被海浪拍打，水花在阳光下闪耀 |
| 103 | A star-filled night sky, with distant snow-covered mountains clearly visible under the moonlight. | 夜空中点缀着繁星，远处的雪山在月光下显得格外清晰 |
| 104 | A brush gliding across the paper, leaving smooth flowing lines. | 一只毛笔正在纸上留下流畅的线条 |
| 105 | A desk with ink and sheets of paper, accompanied by several brushes lying beside them. | 桌面上摆放着墨汁和纸张，旁边放着几支笔 |
| 106 | An artist practicing strokes on paper with steady hand movements and full concentration. | 艺术家专注地在纸上练习笔画，手部动作稳定 |
| 107 | A writing scene under soft lighting, with black lines gradually appearing on the paper. | 灯光下的书写场景，纸张上逐渐显现出黑色的线条 |
| 108 | Several completed writing pieces laid flat on a table, with lines varying in thickness and balance. | 几张完成的书写作品平铺在桌面上，线条粗细有致 |
| 109 | A dancer spinning at the center of the stage, illuminated by overhead lights. | 一名舞者在舞台中央旋转，灯光从上方打下 |
| 110 | An orchestra performing in a concert hall, with the audience quietly listening. | 音乐厅里，乐队正在演奏，观众安静聆听 |
| 111 | The curtain rises on a theater stage as actors begin their performance. | 剧院舞台上拉开了幕布，演员们正在表演 |
| 112 | A solo performer playing the violin under a bright spotlight. | 一名独奏者在聚光灯下演奏小提琴 |
| 113 | A group of dancers performing a synchronized routine on stage. | 舞台上有一群舞者在整齐地表演群舞 |
| 114 | A live music performance on an outdoor stage, with the audience waving glow sticks. | 户外舞台上正在进行一场音乐演出，观众举着灯光棒 |
| 115 | A singer performing at the center of the stage, with a large screen behind them. | 舞台中央有一名歌手在演唱，背后是大型屏幕 |
| 116 | In a contemporary dance performance, dancers use body movements to convey emotions. | 舞蹈表演中，舞者们利用肢体语言表达情感 |
| 117 | Band members on stage playing different instruments in perfect coordination. | 乐队成员在舞台上演奏不同的乐器，配合默契 |
| 118 | Stage lights changing rhythmically in sync with the music during a performance. | 舞台上的灯光随着音乐节奏不断变换 |
| 119 | A model is presenting an elegant outfit on display. | 一名模特正在展示一套优雅的服装 |

| ID | English prompt | Chinese prompt |
| --- | --- | --- |
| 120 | A designer in the studio making final adjustments to a garment. | 一位设计师在工作室里为衣服做最后的修改 |
| 121 | Several garments of different styles displayed in a shop window. | 橱窗里摆放着几件不同款式的衣服 |
| 122 | A group of people walking the runway during a fashion show. | 一群人正在时装秀的舞台上走秀 |
| 123 | A student trying on new clothes and adjusting them while looking in the mirror. | 一名学生正在试穿一件新衣服，照着镜子整理 |
| 124 | A marketplace stall filled with colorful clothes hanging on display. | 市场的摊位上挂满了色彩鲜艳的衣服 |
| 125 | Clothing neatly arranged on hangers in different sizes. | 衣架上整齐地排列着不同尺码的服装 |
| 126 | A tailor taking measurements to make custom-fitted clothing for a client. | 一名裁缝正在为顾客量身定制衣服 |
| 127 | An actor backstage changing outfits in preparation for a performance. | 一位演员在后台换装准备演出 |
| 128 | A rack in a laundry shop filled with freshly cleaned clothes. | 洗衣店的架子上堆放着刚清洗好的衣服 |
| 129 | A couple looking at each other during a wedding ceremony, surrounded by guests as witnesses. | 一对新人在婚礼上互相对视，周围有宾客见证 |
| 130 | A wedding hall decorated with flowers and rows of chairs, awaiting the start of the ceremony. | 婚礼大厅里布置着鲜花和长椅，等待仪式开始 |
| 131 | A couple holding hands and walking toward the center of the stage during the ceremony. | 一对新人在仪式上牵着手走向舞台中央 |
| 132 | Guests clapping and celebrating at the wedding venue. | 宾客们在婚礼现场鼓掌庆祝 |
| 133 | After the wedding, the couple takes group photos with family and friends. | 婚礼结束后，新人和亲友们一起合影 |
| 134 | The couple exchanging vows under an arched decoration. | 新人站在拱形装饰下交换誓言 |
| 135 | A child scattering flower petals along the wedding aisle. | 一个孩子在婚礼过道上撒花瓣 |
| 136 | On the wedding dance floor, the couple begins their first dance. | 婚礼现场的舞池上，新人开始了第一支舞 |
| 137 | Guests seated around decorated tables, waiting for the wedding banquet to begin. | 宾客们围坐在布置好的餐桌旁，等待婚宴开始 |
| 138 | The wedding venue illuminated at night, creating a romantic atmosphere. | 夜晚的婚礼场地被灯光点亮，营造出浪漫的氛围 |
| 139 | A living room with a sofa and a coffee table, sunlight streaming through the window. | 一个客厅里摆放着沙发和茶几，窗外透进阳光 |
| 140 | A bedroom with a neatly made bed and a small lamp on the bedside table. | 卧室里有一张整齐的床，床头放着一盏小灯 |
| 141 | A study room with bookshelves filled with books, and a desk holding a computer and notebook. | 书房的书架上摆满了书，桌上放着电脑和笔记本 |
| 142 | A dining room with the table set with dishes and cutlery, ready for a family meal. | 餐厅的餐桌上摆好了餐具，等待家人用餐 |
| 143 | A kitchen with neatly arranged utensils and a stove in use. | 厨房里有整齐排列的厨具和正在加热的炉灶 |
| 144 | A balcony with several green plants and a small chair placed beside them. | 阳台上放着几盆绿色植物，旁边有一张小椅子 |
| 145 | A bathroom mirror shining under the lights, with a clean and tidy sink. | 浴室的镜子在灯光下明亮清晰，洗手池干净整洁 |

| ID | English prompt | Chinese prompt |
| --- | --- | --- |
| 146 | A children's room with a small bed and toys scattered on the floor. | 儿童房里有一张小床和几件玩具散落在地上 |
| 147 | Family members sitting together on the living room sofa, engaged in conversation. | 家庭成员围坐在客厅的沙发上聊天 |
| 148 | A staircase area decorated with a few paintings, with clean and bright walls. | 楼梯间挂着几幅装饰画，墙壁干净明亮 |
| 149 | An open-plan office with several employees working at their computers. | 开放式办公室里，几名员工正在电脑前工作 |
| 150 | In a meeting room, a presenter is showing slides to colleagues. | 会议室里，一位讲解者正在向同事展示幻灯片 |
| 151 | A desk piled with documents and a laptop computer. | 办公桌上堆放着文件和笔记本电脑 |
| 152 | Employees chatting in the break room next to a coffee machine. | 员工们在茶水间交流，旁边有咖啡机 |
| 153 | An engineer adjusting equipment in a laboratory. | 一名工程师在实验室里调试设备 |
| 154 | A company lobby with a reception desk and visitors waiting. | 公司大厅里有接待台和等待的来访者 |
| 155 | The office windows offer a view of the city skyline. | 办公室的窗外可以看到城市天际线 |
| 156 | An employee retrieving documents beside a printer. | 一名员工正在打印机旁取文件 |
| 157 | Several people discussing project progress in front of a whiteboard. | 几个人在白板前讨论项目进展 |
| 158 | Lights still on in the office at night, showing that someone is working late. | 夜晚的办公室里仍有灯光亮着，显示有人加班 |
| 159 | A young person selecting fruits in a supermarket. | 一名年轻人正在超市里挑选水果 |
| 160 | Someone reading a book on a bench in the park. | 有人在公园的长椅上看书 |
| 161 | A man washing dishes in the kitchen. | 一位男子在厨房里洗碗 |
| 162 | Two people chatting at a street-side café. | 两个人在街头咖啡店聊天 |
| 163 | A mother helping her child with homework. | 一位母亲正在帮助孩子做家庭作业 |
| 164 | A cyclist riding along a quiet street. | 一名骑自行车的人经过安静的街道 |
| 165 | Someone watering plants on a balcony. | 有人在阳台上给植物浇水 |
| 166 | A family watching television together in the living room. | 一家人在客厅里一起看电视 |
| 167 | A young person exercising on a treadmill at the gym. | 一位年轻人在健身房里跑步机上锻炼 |
| 168 | Several people quietly looking at their phones on public transport. | 几个人在公共交通上安静地看手机 |
| 169 | An ancient stone wall surrounding a courtyard, its surface marked by the passage of time. | 一座古老的石墙围绕着庭院，墙面已经布满岁月的痕迹 |
| 170 | A tall ancient tower standing on a hillside, with each level narrowing upward. | 高塔屹立在山坡上，塔身逐层向上收窄 |
| 171 | An old stone bridge spanning across a river, with a pathway paved with blocks of stone. | 一条古桥横跨在河流上，桥面由石块铺成 |
| 172 | An ancient arched gateway standing in the center of a plaza. | 广场中央耸立着一座古代拱门 |
| 173 | Ruins of an ancient site with rows of remaining columns. | 一片古代遗迹，残留的柱子整齐排列 |
| 174 | An old stone staircase leading up to a building at the top of a hill. | 古老的阶梯通向山顶上的建筑 |

| ID | English prompt | Chinese prompt |
|---|---|---|
| 175 | An abandoned castle standing on the grassland, with parts of its walls collapsed. | 废弃的城堡矗立在草原上，部分墙体已坍塌 |
| 176 | The roof of an ancient building covered with neatly arranged tiles. | 古建筑的屋顶覆盖着整齐的瓦片 |
| 177 | A long corridor supported by stone pillars, stretching out on both sides. | 一条长长的走廊由石柱支撑，两侧延伸开去 |
| 178 | An ancient ruin located in a valley, with stone structures faintly visible. | 山谷间有一处古代遗址，石头结构依稀可见 |
| 179 | A plate of fresh fruit neatly sliced and arranged. | 一盘新鲜的水果被整齐地切片摆放在盘子里 |
| 180 | A steaming bowl of soup placed on the dining table. | 餐桌上放着一碗热气腾腾的汤 |
| 181 | A chef chopping vegetables in the kitchen. | 一名厨师正在厨房里切蔬菜 |
| 182 | Several glasses of drinks neatly arranged on a tray. | 几杯饮料整齐地摆放在托盘上 |
| 183 | A market stall filled with a variety of fresh ingredients. | 市场的摊位上摆满了各种新鲜食材 |
| 184 | Several homemade dishes arranged on a dining table, ready to be served. | 餐桌上摆放着几道家常菜，正在等待开餐 |
| 185 | A waiter carrying a tray of food toward the guests. | 一名服务员端着托盘走向客人 |
| 186 | A transparent glass filled with ice cubes and a refreshing drink. | 透明玻璃杯中盛满了冰块和饮品 |
| 187 | A large crowd gathered in the square, with stage lights illuminating the night. | 广场上聚集了许多人，舞台灯光照亮了整个夜晚 |
| 188 | Fireworks exploding in the night sky, lighting up the smiling faces of the crowd. | 夜空中绽放的烟花映照出人群的笑脸 |
| 189 | Colorful lights hanging along both sides of the street, creating a lively atmosphere. | 街道两旁挂满了彩灯，营造出热闹的氛围 |
| 190 | A group of children playing games at a festive gathering. | 一群孩子在节日集会上玩游戏 |
| 191 | A singer performing on a festival stage, with the audience waving glow sticks. | 节日的舞台上有歌手在表演，观众挥舞着手中的灯光棒 |
| 192 | Festival stalls filled with snacks, with people lining up to buy them. | 摊位上摆满了节日小吃，人们排队购买 |
| 193 | Floats slowly moving down the street during a festival parade. | 节日游行中，花车缓缓驶过街道 |
| 194 | People dancing together in the center of the square to celebrate. | 人们在广场中央一起跳舞庆祝 |
| 195 | The festival venue at night decorated with colorful lights, creating a vibrant scene. | 夜晚的节日场地被五彩灯光装点得绚丽多彩 |
| 196 | A group of friends taking photos together at a festive party. | 一群朋友在节日派对上合影留念 |
| 197 | An artist working on a canvas, creating a new artwork. | 一名艺术家正在画布上创作作品 |
| 198 | A gallery displaying several paintings hanging on the walls. | 画廊里展示着几幅挂在墙上的绘画 |
| 199 | Visitors quietly observing an art installation in an exhibition hall. | 观众在展览厅里安静地欣赏艺术装置 |

## Architecture

EN: A modern skyscraper with reflective glass windows standing tall against a clear blue sky
CN: 一座现代化的摩天大楼，玻璃窗反射着清澈的蓝天

EN: A cozy cottage surrounded by a lush garden, with a small porch and a white picket fence
CN: 一座温馨的小屋，四周是郁郁葱葱的花园，门前有一个小露台和白色栅栏

EN: A large, open-plan office with sleek furniture, high ceilings, and plenty of natural light
CN: 一间宽敞的开放式办公室，配有简洁的家具，高高的天花板和充足的自然光

EN: A traditional stone house with a thatched roof nestled in a rural setting
CN: 一座传统的石屋，茅草屋顶，坐落在乡村环境中

EN: A luxurious hotel lobby with grand chandeliers, marble floors, and elegant furniture
CN: 一座豪华酒店的大堂，挂着宏伟的吊灯，大理石地板和优雅的家具

EN: A quaint coffee shop with vintage décor, wooden tables, and cozy seating arrangements
CN: 一个别致的咖啡店，复古风格的装饰，木制桌子和舒适的座位

EN: An old brick building with ivy climbing up its walls, located in a historic district
CN: 一座古老的砖砌建筑，爬满了常春藤，位于历史街区

EN: A minimalist home with clean lines, large windows, and an open-concept living area
CN: 一座极简主义风格的家，简洁的线条，大窗户和开放式客厅

EN: A public library with rows of bookshelves, reading tables, and large windows letting in natural light
CN: 一个公共图书馆，书架整齐排列，设有阅读桌，窗外自然光洒入

EN: A classical building with ornate columns, arches, and sculptures
CN: 一座典雅的古典风格建筑，装饰精美的柱子、拱门和雕塑

EN: A fire station with red garage doors and a tall watchtower
CN: 一座消防站，有红色车库门和高高的瞭望塔

EN: A cozy mountain cabin with a slanted roof, warm lights, and snow on the ground
CN: 一间温馨的山间木屋，屋顶倾斜，灯光暖黄，地上覆盖着积雪

EN: An indoor sports arena with a curved roof, stadium seating, and a basketball court
CN: 一座室内体育馆，有弯曲的屋顶、阶梯看台和篮球场

EN: A suburban house with a neatly trimmed lawn, two-car garage, and a red front door
CN: 一栋郊区住宅，草坪修剪整齐，有双车车库和红色前门

EN: An underground metro station with bright lighting, tiled walls, and directional signs
CN: 一个地铁站，灯光明亮，瓷砖墙面，挂着指示牌

EN: A greenhouse filled with plants, with glass walls and sunlight streaming in
CN: 一座玻璃温室，里面种满了植物，阳光透过玻璃洒满屋内

EN: A beach house on wooden stilts, facing the ocean, with a wide deck and hammock
CN: 一座建在木桩上的海边别墅，面朝大海，有宽敞的平台和吊床

EN: A high-tech research lab with large monitors, lab equipment, and clean white interiors
CN: 一个高科技研究实验室，配有大显示器、实验设备和洁白的内部环境

EN: A rustic barn with wooden beams and a large sliding door, surrounded by open fields
CN: 一座乡村谷仓，有粗大的木梁和大滑门，四周是一望无际的田野

EN: A mountain lodge made of stone and timber, with smoke rising from the chimney
CN: 一间由石材和木材建成的山间小屋，烟囱里升起袅袅炊烟

EN: An art museum with high ceilings, white walls, and minimalist display rooms
CN: 一座艺术博物馆，天花板高高，白墙素净，展厅简洁现代

EN: An airport terminal with large glass facades, rows of check-in counters, and busy travelers
CN: 一个机场航站楼，玻璃幕墙高大透明，排满了值机柜台，人来人往

EN: A rooftop garden on a city building, with plants, benches, and a view of the skyline
CN: 一座城市建筑的屋顶花园，有植物、长椅和城市天际线的美景

EN: A university campus with red brick buildings, clock tower, and green lawns
CN: 一座大学校园，有红砖建筑、钟楼和绿色草坪

EN: A lakeside villa with large glass doors opening to a wooden dock
CN: 一栋湖边别墅，有大玻璃门通向木制码头

Figure 19: Examples of text prompts from the *Architecture* category used for testing cultural neutrality CLIP score. Each pair includes an English prompt and its corresponding Chinese translation. These prompts are designed to avoid cultural or regional bias while maintaining scene diversity and clarity.

---

**Food**

EN: A plate of freshly cooked pasta with tomato sauce.
CN: 一盘新鲜烹饪的面，配有番茄酱

EN: A colorful salad with mixed greens, cherry tomatoes, cucumbers, and a vinaigrette dressing
CN: 一份色彩丰富的沙拉，包含混合生菜、樱桃番茄、黄瓜和香醋沙拉酱

EN: A hot bowl of soup with chunks of vegetables, meat, and noodles, served in a rustic ceramic bowl
CN: 一碗热气腾腾的汤，里面有蔬菜块、肉块和面条，盛在古朴的陶瓷碗中

EN: A basket of freshly baked bread rolls with a side of butter and jam
CN: 一篮新鲜出炉的面包卷，配有黄油和果酱

EN: A sushi platter with assorted pieces of nigiri, sashimi, and rolls, served with soy sauce and wasabi
CN: 一盘寿司，包含各种生鱼片、寿司卷和手握寿司，配有酱油和芥末

EN: A steaming bowl of rice topped with stir-fried vegetables and a fried egg
CN: 一碗热腾腾的米饭，上面放着炒蔬菜和煎蛋

EN: A slice of chocolate cake with layers of cream and a rich ganache topping
CN: 一块巧克力蛋糕，层层奶油和浓郁的甘纳许酱

EN: A smoothie bowl topped with granola, fresh fruits, and a drizzle of honey
CN: 一碗冰沙，顶部撒上格兰诺拉麦片、新鲜水果和一丝蜂蜜

EN: A plate of grilled chicken with a side of roasted vegetables and mashed potatoes
CN: 一盘烤鸡肉，配有烤蔬菜和土豆泥

EN: A traditional breakfast with scrambled eggs, bacon, toast, and a cup of freshly brewed coffee
CN: 一份传统早餐，包括炒蛋、培根、吐司和一杯新鲜冲泡的咖啡

EN: A fresh garden salad with leafy greens, tomatoes, and a light dressing
CN: 一份新鲜的蔬菜沙拉，包含生菜、番茄和清淡调味汁

EN: A bowl of hot soup with mixed vegetables and herbs, steaming gently
CN: 一碗热腾腾的汤，混合蔬菜和香草，轻轻冒着热气

EN: A simple sandwich filled with fresh vegetables and cheese
CN: 一个夹着新鲜蔬菜和奶酪的简单三明治

EN: A glass of fruit smoothie served with a straw and ice cubes
CN: 一杯果味冰沙，配有吸管和冰块

EN: A baked pastry on a white plate, with golden crust and soft filling
CN: 一块烘焙点心，金黄的外壳包裹着柔软的内馅，放在白色盘子上

EN: A cup of hot beverage beside a plate of small biscuits
CN: 一杯热饮放在装有小饼干的盘子旁

EN: A fruit bowl with apples, bananas, and grapes on a kitchen counter
CN: 厨房台面上的一个水果碗，里面有苹果、香蕉和葡萄

EN: A bowl of rice served with a side of cooked vegetables
CN: 一碗米饭配上一份熟蔬菜

EN: A dessert plate with a small cake and a dollop of whipped cream
CN: 甜点盘上有一块小蛋糕和一团鲜奶油

EN: A clear glass of cold water with lemon slices
CN: 一杯透明的柠檬水，杯中漂浮着几片柠檬

EN: A snack tray with nuts, dried fruits, and crackers
CN: 一盘小吃，包含坚果、干果和薄脆饼干

EN: A plain bowl of cereal with milk and sliced bananas
CN: 一碗简单的麦片，加了牛奶和香蕉片

EN: An open-faced toast topped with avocado and a poached egg
CN: 一片吐司上铺着鳄梨和水煮蛋

EN: A plate of roasted vegetables with crispy edges on a wooden table
CN: 一盘边缘微焦的烤蔬菜，摆在木质桌面上

EN: A stack of fluffy pancakes topped with syrup and fruit slices
CN: 一叠松软的煎饼，上面淋着糖浆和水果片

Figure 20: Examples of text prompts from the *Food* category used for testing cultural neutrality CLIP score. Each pair includes an English prompt and its corresponding Chinese translation. These prompts are designed to avoid cultural or regional bias while maintaining scene diversity and clarity.

**Festival**

EN: A parade with a giant dragon and dancers moving through a street decorated with banners and colorful lights
CN: 游行队伍中，巨龙与舞者穿过布满横幅和彩灯的街道

EN: A family gathering around a dinner table during a traditional festival, with dishes shared among the group
CN: 传统节日中，一家人围坐在餐桌旁，共享丰盛的佳肴

EN: A romantic evening with a couple under a full moon, enjoying a picnic and exchanging gifts
CN: 一对情侣在明亮的圆月下享受野餐，互赠礼物，浪漫至极

EN: Children running around with lanterns during a night festival, laughing and playing together
CN: 夜晚的节日中，孩子们手持灯笼，笑着玩耍

EN: Families flying colorful kites on a windy day during a cultural holiday
CN: 文化节日里，家庭成员在风和日丽的日子里放飞色彩缤纷的风筝

EN: A boat race with decorated vessels on a river during a seasonal festival
CN: 节庆期间，装饰华丽的船只在河上竞渡

EN: Elders gathering in the mountains, enjoying tea and nature on a festive day
CN: 节日里，老人们在山间聚集，品茶享受自然风光

EN: A street lined with decorations for a holiday celebration, with families and friends walking together
CN: 节日庆典时，街道两旁装饰着彩带，家庭和朋友们并肩走过

EN: A night festival with lanterns floating on a river, illuminating the water with soft light
CN: 夜晚的节日，灯笼漂浮在河面上，温柔的光辉照亮水面

EN: People enjoying a public celebration, watching fireworks and eating traditional snacks
CN: 人们在公开庆典中，观看烟花表演，享受传统小吃

EN: A rooftop party with hanging lights and people chatting under the stars
CN: 屋顶派对上悬挂着灯串，人们在星空下交谈

EN: A group of people lighting candles together in a peaceful evening ceremony
CN: 一群人在宁静的傍晚共同点亮蜡烛，气氛祥和

EN: People gathering in a city square for a public celebration with lights and music
CN: 人们聚集在城市广场上，灯光与音乐营造出热闹的庆典氛围

EN: A group of friends watching fireworks together under a night sky
CN: 一群朋友在夜空下仰望烟花绽放

EN: Families enjoying a picnic in the park with balloons and festive decorations
CN: 家庭成员在公园里野餐，四周布置着气球和节日装饰

EN: Children running through a field during a festive outdoor event
CN: 节日活动中，孩子们在草地上欢快奔跑

EN: A quiet street illuminated with string lights and festive colors
CN: 安静的街道上挂满了彩灯和节日色彩装饰

EN: People dancing in a parade with colorful costumes and cheerful music
CN: 游行中人们穿着鲜艳服饰伴随欢乐音乐跳舞

EN: A decorated table with candles, desserts, and drinks during a celebration
CN: 庆典中摆满甜点、饮品与蜡烛的装饰餐桌

EN: A family taking photos together in front of a decorated house
CN: 一家人在装饰过的房屋前合影留念

EN: People walking along a waterfront at night during a festival of lights
CN: 节日灯光秀中，人们夜晚漫步在水边

EN: Couples enjoying a romantic dinner with dim lighting and soft music
CN: 情侣在昏黄灯光和柔和音乐中共进浪漫晚餐

EN: Kids playing with glow sticks and bubbles at an evening gathering
CN: 孩子们在夜晚聚会中玩耍荧光棒和泡泡

EN: People sitting on a hillside watching a light show in the distance
CN: 人们坐在山坡上远望灯光秀

EN: A neighborhood organizing a street fair with food stalls and games
CN: 街区举行嘉年华，设有美食摊位与游乐项目

Figure 21: Examples of text prompts from the *Festival* category used for testing cultural neutrality CLIP score. Each pair includes an English prompt and its corresponding Chinese translation. These prompts are designed to avoid cultural or regional bias while maintaining scene diversity and clarity.

## Art

EN: An artist is working on a canvas, adding details with focused attention.
CN: 一名艺术家正在画布上创作

EN: A gallery space with several paintings on display, arranged neatly along the walls.
CN: 画廊中摆放着几幅正在展出的绘画作品

EN: A sculptor in a studio carefully shaping a stone piece with tools.
CN: 一名雕塑家在工作室里打磨石头作品

EN: Visitors in the main hall of an art museum, observing the artworks displayed around them.
CN: 艺术博物馆的大厅里，观众正在欣赏展出的作品

EN: An artist mixing different colors on a palette at a worktable.
CN: 一名艺术家在工作台上调配颜色

EN: A sculptor carving a block of stone with tools.
CN: 一名雕塑家正在用工具雕刻石块

EN: A photographer adjusting the camera to shoot a model.
CN: 摄影师正在调整相机对着模特拍摄

EN: Several ceramic artworks displayed inside the exhibition case.
CN: 展览柜里展示着几件陶瓷艺术品

EN: Actors rehearsing a performance on stage.
CN: 舞台上的演员正在排练一场表演

EN: An art studio table filled with brushes and paints.
CN: 艺术工作室的桌子上放满了画笔和颜料

EN: A student copying an oil painting.
CN: 一名学生正在临摹一幅油画

EN: Sculptures of various materials displayed in the hall of a sculpture exhibition.
CN: 雕塑展的大厅中摆放着不同材质的雕塑

EN: A huge black-and-white photograph hanging in the exhibition hall.
CN: 展厅里悬挂着一张巨大的黑白摄影作品

EN: A painter adding details to a canvas in the studio.
CN: 画家正在工作室里为画布添加细节

EN: A sculptor polishing the surface of a statue.
CN: 雕塑家正在为一尊雕像打磨表面

EN: Visitors viewing black-and-white photographs at a photo exhibition.
CN: 摄影展上观众正在欣赏黑白照片

EN: Dancers practicing movements on the stage.
CN: 舞蹈演员在舞台上练习排练动作

EN: An artist creating graffiti artwork outdoors.
CN: 艺术家正在户外进行涂鸦创作

EN: Stage lights focusing on a solo performer.
CN: 舞台上灯光照射在独唱演员身上

EN: Visitors lining up to enter the art exhibition hall.
CN: 观众排队进入艺术展览的大厅

EN: A model posing for a painter in the studio.
CN: 一位模特在工作室里为画家摆姿势

EN: The stage curtain slowly rising as the performance is about to begin.
CN: 舞台上的幕布缓缓升起，表演即将开始

EN: Students sitting together sketching in an art classroom.
CN: 美术教室里学生们围坐在一起画素描

EN: Several watercolor paintings displayed at the art exhibition.
CN: 画展上摆放着几幅水彩画

EN: A designer drawing fashion sketches on the desk.
CN: 设计师正在桌上绘制时装草图

Figure 22: Examples of text prompts from the *Art* category used for testing cultural neutrality CLIP score. Each pair includes an English prompt and its corresponding Chinese translation. These prompts are designed to avoid cultural or regional bias while maintaining scene diversity and clarity.

## Clothing

EN: A model is presenting an elegant outfit on display.
CN: 一名模特正在展示一套优雅的服装

EN: A designer in the studio making final adjustments to a garment.
CN: 一位设计师在工作室里为衣服做最后的修改

EN: Several garments of different styles displayed in a shop window.
CN: 橱窗里摆放着几件不同款式的衣服

EN: A group of people walking the runway during a fashion show.
CN: 一群人正在时装秀的舞台上走秀

EN: A student trying on new clothes and adjusting them while looking in the mirror.
CN: 一名学生正在试穿一件新衣服，照着镜子整理

EN: A marketplace stall filled with colorful clothes hanging on display.
CN: 市场的摊位上挂满了色彩鲜艳的衣服

EN: Clothing neatly arranged on hangers in different sizes.
CN: 衣架上整齐地排列着不同尺码的服装

EN: A tailor taking measurements to make custom-fitted clothing for a client.
CN: 一名裁缝正在为顾客量身定制衣服

EN: An actor backstage changing outfits in preparation for a performance.
CN: 一位演员在后台换装准备演出

EN: A rack in a laundry shop filled with freshly cleaned clothes.
CN: 洗衣店的架子上堆放着刚清洗好的衣服

EN: A model walking on stage in a long dress during a fashion show.
CN: 一名模特穿着长裙在舞台上走秀

EN: A shop window displaying a variety of coats.
CN: 商店橱窗里展示着各式各样的外套

EN: A woman in a raincoat walking with an umbrella on the wet sidewalk.
CN: 一名女子穿着雨衣打着伞走过湿漉漉的人行道

EN: A man jogging in the park wearing sportswear.
CN: 一位男子身穿运动服在公园里跑步

EN: Clothing racks neatly filled with shirts in different colors.
CN: 衣架上整齐挂满了不同颜色的衬衫

EN: A woman walking down the street in a hooded jacket.
CN: 一位女子穿着连帽外套在街头行走

EN: Models lined up to showcase outfits at a fashion show.
CN: 时装发布会上模特们排成一列展示服饰

EN: A worker wearing a uniform while working in a workshop.
CN: 一名工人穿着制服正在车间里工作

EN: Brightly colored summer dresses displayed in a shopping mall.
CN: 商场里展示着色彩鲜艳的夏季连衣裙

EN: A young person strolling in the street wearing jeans and a loose jacket.
CN: 一位年轻人穿着牛仔裤和宽松外套在街头闲逛

EN: Several players warming up in sports jerseys by the field.
CN: 运动场边，几名球员穿着球衣正在热身

EN: At a wedding, the couple walking toward the stage in formal attire.
CN: 婚礼现场，新人身穿礼服走向舞台

EN: Sweaters and scarves neatly folded inside a wardrobe.
CN: 衣柜里整齐叠放着毛衣和围巾

EN: A designer fitting new clothes on a model.
CN: 一位设计师正在为模特试穿新衣服

EN: A clothing store entrance on the street with discount signs and hanging racks.
CN: 街头的服装店门口挂着打折广告和衣架

Figure 23: Examples of text prompts from the *Clothing* category used for testing cultural neutrality CLIP score. Each pair includes an English prompt and its corresponding Chinese translation. These prompts are designed to avoid cultural or regional bias while maintaining scene diversity and clarity.

