# OpenReview forum: "CTA-Flux: Integrating Chinese Cultural Semantics into High-Quality English Text-to-Image Communities"
_ICLR.cc/2026/Conference — Submitted to ICLR 2026_

### Official Review · Reviewer_Ly1r · 2025-10-31

**Soundness:** 3
**Presentation:** 3
**Contribution:** 3
**Rating:** 6
**Confidence:** 2

**Summary:**

This paper addresses the challenge of extending English-centric text-to-image (T2I) models to support Chinese language prompts while preserving cultural semantics. The authors propose CTA-Flux, a lightweight bilingual extension of the Flux model, which introduces a Chinese Linguistic Attention Branch (CLAB) and a representation alignment loss to bridge linguistic and visual distribution gaps between English and Chinese prompts. Quantitative and qualitative experiments suggest that CTA-Flux achieves comparable or better performance than Flux on English benchmarks, while significantly improving cultural faithfulness for Chinese prompts.

**Strengths:**

1. Cross-lingual and cross-cultural alignment in generative models is underexplored yet increasingly important.
2. The approach extends an existing large English T2I model without retraining it from scratch, which is pragmatic for deployment.
3. Maintaining plugin compatibility (LoRA, ControlNet) is an appealing engineering consideration for community adoption.
4. The authors provide both quantitative (FID, CLIP score, GenEval) and qualitative (human cultural authenticity ratings) results to support claims.

**Weaknesses:**

1. The method is presented as generalizable to other non-English languages, but experiments focus exclusively on Chinese. It remains unclear whether the proposed architecture and training scheme would generalize effectively to languages with very different morphology or script systems.
2. While the paper aims to improve cultural authenticity, it does not discuss or measure potential cultural stereotyping or bias amplification, which are crucial ethical aspects of “cultural-aware” generation.
3. The cultural-specific prompt set appears hand-crafted and may not represent the full diversity of Chinese culture or real user prompts. A more systematic benchmark or open-sourced dataset would strengthen reproducibility.
4. The architectural idea (auxiliary language branch + feature alignment) resembles existing multimodal adaptation techniques; the contribution lies more in application and empirical validation than in theoretical innovation.

**Questions:**

No

---

> ### Author Response · Authors · 2025-11-21
> **Response to Reviewer Ly1r (Part 1/2)**
>
> **General Reply**
>
> Thank you very much for your valuable comments. Below, we provide detailed responses to each of your questions and comments. If you have any further questions or concerns regarding our paper or our responses, please let us know, and we will address them promptly.
>
> **W1: Generalization to other languages beyond Chinese.**
>
> > The method is presented as generalizable to other non-English languages, but experiments focus exclusively on Chinese. It remains unclear whether the proposed architecture and training scheme would generalize effectively to languages with very different morphology or script systems.
>
> **A1:** Thank you for your question. We have added Appendix C.6 (Zero-Shot Multilingual Generalization) and Fig. 14 to the revised paper.
>
> - **Reason for choosing Chinese:** We chose Chinese as our first test case because of its significant differences from English in language (e.g., morphology, writing system) and culture (e.g., visual distribution), making it a highly challenging "stress test."
> - **Generalization potential of the framework:** Our CTA-Flux framework is designed to be language-agnostic. Its core mechanism (as shown in Figure 1) is: (a) freezing the English backbone (T5 + MMDiT); (b) introducing a language-specific text encoder (Text Encoder CN, we used Qwen); (c) aligning the language with the feature space of English T5 through a lightweight adapter and a $\mathcal{L}_{RA}$ representation alignment loss.
> - **Extending to other languages:** To extend it to other languages, simply replace "Text Encoder CN" with the corresponding state-of-the-art encoder for that language and retrain the lightweight CLAB adapter using our two-stage training strategy (specifically, $\mathcal{L}_{RA}$). The MMDiT backbone requires no modifications.
> - **Zero-Shot Multilingual Generalization:** Although our model was trained only on Chinese-English pairs, we evaluated it on Spanish, French, Russian, and Italian. As shown in Fig. 14, CTA-Flux successfully generates semantically accurate images for these unseen languages. This generalization capability stems from our Stage-1 semantic alignment strategy (Section 4.2). By aligning the non-English language encoder (Qwen) with the English backbone's semantic space, the model learns a language-agnostic conceptual representation rather than simple token mapping(). Since the Qwen encoder inherently supports multiple languages, this alignment transfers to other languages zero-shot. This demonstrates the multilingual potential of CTA-Flux. Its effectiveness will be further validated in the future by training with data in other languages.
>
> **W2: Cultural stereotyping and bias amplification.**
>
> > While the paper aims to improve cultural authenticity, it does not discuss or measure potential cultural stereotyping or bias amplification, which are crucial ethical aspects of “cultural-aware” generation.
>
> **A2:** Thank you for your question and this is a crucial addition. We greatly appreciate you pointing out this key ethical dimension in the generative model of "cultural perception."
>
> - **First,** our two-stage training strategy mitigates this problem to some extent. The first stage (general alignment) uses billions of diverse general image-text pairs (mixed Chinese and English), which helps the model learn "general semantics" rather than "stereotypes." The second stage (cultural fine-tuning), while using 40,000 culturally specific data pairs, strives to reduce undesirable biases through rigorous data filtering (e.g., excluding low aesthetic appeal or weak alignment).
> - **Then,** Our cultural tendency test (Table 2) aims to measure whether the model's generated results are more consistent with the data distribution of Chinese culture (e.g., Asian faces) when faced with neutral cues (e.g., "a person"), validating the model's cultural authenticity. This is different from generating harmful, oversimplified stereotypes (e.g., all Chinese people wear specific clothing).
> - We added Appendix C.5 and Fig. 13 of our revised paper to evaluate whether the model introduces cultural stereotypes or engages in simplistic visual collage when the cue words do not contain explicit cultural cues. As shown in Fig. 13, our model does not inject unnecessary Chinese cultural symbols or pile up irrelevant visual elements, for example, the two Chinese individuals in the figure are dressed in casual skirts and a business suit rather than traditional attire, resulting in natural and realistic imagery.
>
> So, these results demonstrate that our model avoids generative behavior with cultural stereotypes or collage-like patterns, generating coherent, context-appropriate, and high-fidelity images even without explicitly provided cultural semantics.

---

> ### Author Response · Authors · 2025-11-21
> **Response to Reviewer Ly1r (Part 2/2)**
>
> **W3: Dataset limitations and benchmarking.**
>
> > The cultural-specific prompt set appears hand-crafted and may not represent the full diversity of Chinese culture or real user prompts. A more systematic benchmark or open-sourced dataset would strengthen reproducibility.
>
> **A3:** Thank you for your question. We address the evaluation comprehensiveness in two ways:
>
> - **Standardized Benchmarks:** To ensure we are not relying solely on hand-crafted prompts, we evaluated our model on four established rigorous benchmarks: MS-COCO30K, GenEval(Section 5.2), T2I-CompBench++ and DPG-Bench (Appendix A). Results show that CTA-Flux maintains competitive performance with Flux on complex compositional reasoning tasks.
> - **Diversity of Our Dataset:** For the cultural evaluation (where no standard benchmark exists for Flux-based Chinese adaptation), we curated a dataset balanced across 15 distinct categories (e.g., Architecture, Food, Festivals, Daily Life), as visualized in Figure 15. This ensures coverage of diverse semantic regions rather than a narrow subset.
> - **Open Source****:** We commit to releasing this benchmark dataset and our evaluation code as well as the model implementation and pretrained weights to the community to facilitate reproducible research in multilingual T2I generation.
>
> **W4: Novelty and contribution type.**
>
> > The architectural idea (auxiliary language branch + feature alignment) resembles existing multimodal adaptation techniques; the contribution lies more in application and empirical validation than in theoretical innovation.
>
> **A4:** Thank you for your question. While adapter-style methods exist, CTA-Flux’s contributions are distinct in three concrete ways:
>
> - This is the first native integration of a non-English branch into Flux’s MMDiT flow-matching backbone (not Unet), enabling direct control of flow generation and achieving a Chinese text-to-image model with performance comparable to Flux. Other adapter architectures are based on image modality information injection, which is different from our language modality injection and the use of empty text in the backbone to share the latent space.
> - A Representation Alignment Loss tailored to align features between encoders while preserving language-specific nuances (we include a thresholding mechanism to avoid over-alignment)
> - We employ a two-stage training approach (first performing shared semantic alignment, then culture-specific fine-tuning) to significantly improve cultural fidelity while maintaining generative capabilities, as demonstrated by extensive quantitative and qualitative experiments.

---

### Official Review · Reviewer_jj2X · 2025-10-31

**Soundness:** 3
**Presentation:** 4
**Contribution:** 3
**Rating:** 6
**Confidence:** 4

**Summary:**

This paper presents a concise and logically coherent idea, bridging the linguistic gap between Chinese and English, which drives T2I models comprehend cultures across different regions. The overall framework leverages language models for different scripts and seamlessly integrates them with MMDiT. The experiments include several image samples generated from Chinese prompts, providing a certain level of illustration for the model’s effectiveness.

**Strengths:**

1. The problem is highly relevant to real-world applications, as the paper focuses on cross-lingual and cross-cultural adaptation, offering significant practical value.

2. The method is designed with simplicity and efficiency in mind. It introduces no changes to the original backbone, preserving full compatibility with community plug-ins such as LoRA.

3. The training strategy is well-designed, employing a two-stage approach. The first stage aligns Chinese and English features, while the second stage concentrates on capturing Chinese cultural semantics, effectively balancing generality with cultural specificity.

4. The approach can, to a certain extent, be generalized to other languages, demonstrating broad applicability.

**Weaknesses:**

1. The definitions and metrics employed for the linguistic and cultural gaps remain somewhat vague, relying heavily on manual evaluation or CLIP-based similarity scores.

2. Although the study proposes metrics to assess the quality of images generated from Chinese prompts, it lacks an evaluation of the depth of Chinese language understanding—such as how well the model handles complex linguistic phenomena like polysemy, idioms, or cultural metaphors.

**Questions:**

1. What is your rationale for choosing MMDiT? Why not opt another backbone to accomplish this task?

2. The Chinese prompts used in the experiments mostly refer to culturally salient items such as festivals, clothing, and food. Could you offer some illustration of prompts that are culturally neutral but linguistically complex?

3. How can the model ensure that it truly understands the Chinese language, rather than merely piecing together stereotypically "Chinese-looking" visual elements?

---

> ### Author Response · Authors · 2025-11-21
> **Response to Reviewer jj2X (Part 1/2)**
>
> **General Reply**
>
> Thank you very much for your valuable comments. Below, we provide detailed responses to each of your questions and comments. If you have any further questions or concerns regarding our paper or our responses, please let us know, and we will address them promptly.
>
> **W1: Definitions and Metrics**
>
> > The definitions and metrics employed for the linguistic and cultural gaps remain somewhat vague, relying heavily on manual evaluation or CLIP-based similarity scores.
>
> **A1:** Thank you for your question. We acknowledge that accurately "defining" and "measuring" cultural and linguistic differences is a challenging open problem in the field. In this work, we followed standard T2I practices to provide a comprehensive evaluation:
>
> - **General Performance (Table 1):** We used FID and CLIP Scores on the MS-COCO benchmark to ensure the model maintains high-quality generation capabilities.
> - **Compositional Capability (Table 1):** We employed GenEval to assess compositional reasoning (e.g., object counting, color attributes), ensuring the model handles complex prompt structures.
> - **Cultural Fidelity (Fig. 3, Fig. 4):** Recognizing that "cultural authenticity" is hard to quantify automatically, we utilized Human Evaluation as the gold standard. We use a test set comprising various categories of culturally specific prompts and culturally neutral prompts for human evaluation, aiming to assess Cultural Fidelity as objectively as possible.  In addition, we provide the detailed human evaluation criteria in Section D of the Appendix and list all prompts used for human evaluation in Section E.
> - **Cultural Tendency (Table 2):** We used CLIP similarity scores on "cultural-neutral" prompts to measure the model's inclination towards Chinese cultural representations.
>
> **W2: Depth of Language Understanding**
>
> > Although the study proposes metrics to assess the quality of images generated from Chinese prompts, it lacks an evaluation of the depth of Chinese language understanding—such as how well the model handles complex linguistic phenomena like polysemy, idioms, or cultural metaphors.
>
> **A2:**  Thank you for your question. In the current version, we explored this through qualitative comparisons in Figure 5:
>
> - **Polysemy Resolution (Fig. 5a):** We demonstrated that while the English word "Crane" is ambiguous (bird vs. machine), CTA-Flux correctly interprets the specific Chinese characters (e.g., "鹤" for bird), resolving ambiguity where translation-based methods fail.
> - **Cultural Concepts (Fig. 5b, 5c):** The model successfully generates specific concepts like "Dim sum" and "Lanterns" better than baselines.
>
> To further reinforce this point, we have added more cases in the Appendix C.5 for a qualitative evaluation of complex linguistic phenomena, including:
>
> - **Chinese idioms (Fig.12):** e.g., 车水马龙 (Cars, water, horses, dragons). CTA-Flux generates an image of a busy traffic street because it understands this is a Chinese idiom describing heavy road traffic, not an actual scene with horses and dragons. In contrast, Flux generates a literal interpretation featuring real cars, water, horses, and dragons.
> - **Chinese poetry (Fig.12):** e.g., 小桥流水人家 (Small bridge, flowing water, and houses). CTA-Flux generates a landscape image of a Jiangnan (Southern China) water town because it understands this is a line from Classical Chinese poetry. Conversely, Flux generates a scene with European-style architecture and landscape.
> - **Chinese metaphors (Fig.12):** e.g., 蚂蚁上树 (Ants Climbing a Tree). CTA-Flux generates an image of a Chinese dish because it understands that the word ‘ants’ is **figurative** (referring to the **minced meat**), and not actual ants. However, Flux generates an image that truly features **ants**, which is clearly inconsistent with the reality.

---

> ### Author Response · Authors · 2025-11-21
> **Response to Reviewer jj2X (Part 2/2)**
>
> **Q1: Rationale for Choosing MMDiT**
>
> > What is your rationale for choosing MMDiT? Why not opt another backbone to accomplish this task?
>
> **A3:** Thank you for your question. We selected the Flux backbone (based on MMDiT) for two primary reasons:
>
> - **Forward-Looking Technology:** Flux represents the state-of-the-art in T2I, utilizing Flow Matching and Transformers. Adapting the MMDiT architecture is more valuable to the community than optimizing older U-Net architectures like Stable Diffusion 1.5.
> - **Architectural Fit:** The MMDiT architecture is natively designed for multimodal inputs. Its Double Stream and Single Stream blocks allow us to seamlessly inject Chinese semantics via the Chinese Linguistic Attention Branch (CLAB). This aligns perfectly with our goal of training a lightweight adapter while keeping the massive backbone parameters frozen.
>
> **Q2: Culturally Neutral & Linguistically Complex Prompts**
>
> > The Chinese prompts used in the experiments mostly refer to culturally salient items such as festivals, clothing, and food. Could you offer some illustration of prompts that are culturally neutral but linguistically complex?
>
> **A4:** Thank you for your question. Our main text focused on cultural keywords to verify the resolution of "cultural bias". However, to address your concern regarding structural complexity, we add examples to Appendix Fig. 11, including human scenes and natural scenes, to prove that CTA-Flux can learn culturally neutral but linguistically complex semantics, not merely cultural symbols. More culturally neutral prompts have been added to Table 8 in Appendix E.
>
> **Q3: Understanding vs. Stereotyping**
>
> > How can the model ensure that it truly understands the Chinese language, rather than merely piecing together stereotypically "Chinese-looking" visual elements?
>
> **A5:** Thank you for your question. We ensure "true understanding" through our Two-Stage Training Strategy and Representation Alignment Loss ($\mathcal{L}_{RA}$)
>
> - **Stage 1 (Alignment):** We use a massive dataset (1B pairs, mixed English/Chinese) and $\mathcal{L}_{RA}$ to align the Chinese encoder with the English encoder. This forces the model to learn universal semantics (e.g., mapping the Chinese word for "cat" to the general concept of "cat", not a "Chinese cat").
> - **Stage 2 (Cultural Fine-Tuning):** We strictly limit cultural-specific learning to a smaller, high-quality dataset (40k pairs) in the second stage.
>
> **Evidence:**
>
> - **General Performance (Table 1):** On the general MS-COCO benchmark, CTA-Flux (FID 15.40) outperforms the original Flux (16.39) without degradation. If the model were overfitting to stereotypes, general performance would suffer.
> - **Neutral Scenes (Fig. 4):** In human evaluations of cultural-neutral prompts, our model shows no significant difference in semantic consistency or image quality compared to Flux ($p > 0.05$). This proves it does not force Chinese elements into neutral contexts.
> - **Chinese Culture (Fig. 5, 10, 12):** CTA-Flux demonstrates a genuine understanding of Chinese culture and its nuances—from polysemy and traditional elements (attire, food, festivals) to culturally loaded idioms and metaphors. The generated images precisely match the Chinese linguistic context, confirming that the system truly comprehends the language rather than merely assembling "Chinese-style" visual elements.

---

### Official Review · Reviewer_TPwn · 2025-10-31

**Soundness:** 3
**Presentation:** 3
**Contribution:** 1
**Rating:** 2
**Confidence:** 5

**Summary:**

This paper proposes CTA-Flux, which adapts an English-only text-to-image model to support both Chinese and English.

**Strengths:**

The paper successfully trains a text-to-image model that understands the semantic meaning of Chinese text, enabling Flux to process Chinese prompts effectively.

**Weaknesses:**

This work lacks novelty: understanding Chinese semantics is achieved simply by replacing the text encoder and retraining the model.

**Questions:**

CTA-Flux is a foundational model. Could the authors provide evaluation results on public benchmarks, such as GenEval and T2I-CompBench?

---

> ### Author Response · Authors · 2025-11-21
> **Response to Reviewer TPwn (Part 1/2)**
>
> **General Reply**
>
> Thank you very much for your valuable comments. Below, we provide detailed responses to each of your questions and comments. If you have any further questions or concerns regarding our paper or our responses, please let us know, and we will address them promptly.
>
> **W1. Clarification on Novelty and Methodology**
>
> > This work lacks novelty: understanding Chinese semantics is achieved simply by replacing the text encoder and retraining the model.
>
> **A1:** Thank you for the time spent reviewing. Our method is **not** a simple replacement of the text encoder followed by full retraining. Training a large-scale DiT model from scratch (or fully fine-tuning one) is computationally prohibitive and, more importantly, destroys compatibility with the existing open-source ecosystem.
>
> Our technical contribution is distinct for three key reasons:
>
> - **Frozen Backbone & Community** **Compatibility**: Unlike standard fine-tuning or retraining, we freeze the original Flux backbone and English text encoder (Section 4.1, Section 4.2). We introduce a lightweight Chinese Linguistic Attention Branch (CLAB) that injects Chinese semantics into the MMDiT blocks via a parallel attention mechanism. This architectural choice is critical because it preserves the pre-trained priors of Flux and ensures seamless compatibility with existing community plugins (e.g., ControlNet, LoRA, IP-Adapter), as demonstrated in Figure 6 and Figure 9. A simple "replace and retrain" approach would render all existing community tools incompatible.
> - **Representation Alignment Loss ($\mathcal{L}_{RA}$):** We introduce a novel auxiliary supervision signal (Eq. 3 and 4) to align the feature space of the Chinese encoder (Qwen) with the English encoder (T5). This allows the added branch to leverage the backbone’s existing semantic understanding immediately, accelerating convergence and ensuring semantic fidelity without destabilizing the generation process.
> - **Two-Stage Adaptation Strategy:** We utilize a specific two-stage training pipeline (Section 4.2). Stage 1 aligns the linguistic feature distribution using mixed data, while Stage 2 focuses on the visual feature distribution gap (cultural nuances) using a curated high-quality Chinese dataset. This strategy effectively addresses both linguistic and visual distribution discrepancies in cross-lingual and cross-cultural adaptation, while keeping the original Flux backbone completely frozen.

---

> ### Author Response · Authors · 2025-11-27
> **Response to Reviewer TPwn (Part 2/2)**
>
> **Q1. Evaluation on Public Benchmarks**
>
> > Could the authors provide evaluation results on public benchmarks, such as GenEval and T2I-CompBench?
>
> **A2:** Thank you for your valuable suggestions. We respectfully point out that we have already included the GenEval benchmark results in the initial submitted paper.
>
> - **GenEval Results:** Please refer to Table 1. CTA-Flux achieves an Overall GenEval score of 0.62, which is comparable to the original English-only Flux (0.65). This demonstrates that adding Chinese capability does not degrade the model's complex compositional reasoning capabilities.
> - **T2I-CompBench:** We have added evaluation results on T2I-CompBench in the revised paper, which are presented in Table 5 of the Appendix. The CTA-Flux achieved 35.17 on Complex, which is close to Flux's 36.56. The performance of CTA-Flux is similar to that of Flux across all metrics, and in some cases—such as *Attribute Binding-Color*—our model even shows slight improvements.
> |Method|InputLang.|Color↑|Shape↑|Texture↑|2D-Spatial↑|3D-Spatial↑|Non-Spatial↑|Numeracy↑|Complex↑|
> |---|---|---|---|---|---|---|---|---|---|
> |SD1.5|English|37.58|37.13|41.86|11.65|--|*31.12*|--|30.47|
> |SDXL|English|58.79|46.87|52.99|21.31|35.66|**31.19**|49.91|32.37|
> |Flux|English|*73.79*|**51.96**|**64.64**|**28.50**|**41.62**|30.72|**62.46**|**36.56**|
> |CTA-Flux(Ours)|Chinese&English|**76.70**|*50.24*|*63.44*|*25.16*|*39.22*|30.79|*56.50*|*35.17*|
> - **DPG-Bench:** We have additionally included experimental results on the DPG-Bench test set. The CTA-Flux model posted an Overall score of 80.23, trailing Flux's 83.24 by a small margin. As presented in Table 6, CTA-Flux demonstrates performance comparable to the Flux model across all individual directions and overall, with slight improvements observed in *Entity* dimension.
> |Method↑|InputLang↑|Global↑|Entity↑|Attribute↑|Relation↑|Other↑|Overall↑|
> |---|---|---|---|---|---|---|---|
> |SDv1.5|English|74.63|74.23|75.39|73.49|67.81|63.18|
> |PixArt-α|English|74.97|79.32|78.60|82.57|76.96|71.11|
> |SDXL|English|83.27|82.43|80.91|86.76|80.41|74.65|
> |Flux|English|**89.51**|*85.81*|**88.31**|**91.28**|**90.11**|**83.24**|
> |CTA-Flux(Ours)|Chinese&English|*88.36*|**87.64**|*86.36*|*88.52*|*87.22*|*80.23*|
> - **MS-COCO:** In Table 1, we also provided the standard zero-shot FID-30K and CLIP scores on the MS-COCO dataset. CTA-Flux achieves FID of 15.40, surpassing both SDXL (18.64) and the bilingual BDM (28.34), and performing slightly better than the base Flux (16.39).
>
> We believe these results on GenEval, T2I-CompBench, DPG-Bench, and MS-COCO provide a strong indication of the model’s robustness across public benchmarks.
>
> **Conclusion**
>
> We hope this clarification highlights that CTA-Flux is not a trivial retraining effort but a methodologically designed adaptation framework that solves the “cultural/linguistic gap’’ while preserving “community compatibility’’—a significant challenge in the current Generative AI landscape. Importantly, **a simple replacement of the text encoder followed by retraining cannot address the core problem tackled in this work**, namely enabling a new language and its cultural semantics to be integrated into an English-centric community without sacrificing model or community compatibility.

---

### Author Response · Authors · 2025-11-27
**Summary of Revisions**

Dear Reviewers,

We sincerely thank you for your careful reading of our manuscript and for your constructive and insightful comments. We greatly appreciate your positive feedback on the clarity of our presentation, the motivation of our study, the novelty of the proposed method, and the diversity of our experimental analysis.

In response to your valuable suggestions, we have revised and improved the manuscript accordingly. All major changes have been highlighted in blue in the revised PDF. Below is a brief summary of the main updates:

**1.** Expanded quantitative evaluations on additional public benchmarks to further validate model robustness and general capability. (Appendix A, Table 5-6)

**2.** Added new visual analyses to strengthen the assessment of cultural fidelity and qualitative performance. (Appendix C.3-4, Fig.10-11)

**3.** Introduced deeper analysis of linguistic understanding beyond cultural keywords, including complex language phenomena. (Appendix C.5, Fig.12)

**4.** Supplemented visualization on cultural bias and stereotyping to verify safe and neutral generation behavior. (Appendix C.5, Fig.13)

**5.** Conducted additional experiments on zero-shot multilingual generalization to demonstrate cross-lingual transferability.  (Appendix C.6, Fig.14)

**6.** Provided more detailed descriptions of dataset composition and human evaluation protocols to improve reproducibility. (Appendix D-E, Fig.15-17, Table7-8)

Once again, we thank you for the time and effort you devoted to reviewing our work. **We sincerely hope that the revisions have adequately addressed your comments. We warmly welcome any further questions or discussion and would greatly value your continued feedback.**

---

### Author Response · Authors · 2025-12-03
**Summary Comment (Part 1/2)**

We would like to sincerely thank the reviewers and the Area Chair for their time and efforts. This comment briefly summarizes our main contributions, clarifies a key misunderstanding, and highlights the main updates made in response to the reviews.

### **Summary of Contributions**

We propose a methodological framework that **not only** enables Flux to natively support Chinese text prompts **but also** drives the generated images to align more closely with Chinese linguistic and cultural semantics, thereby mitigating English-centric cultural bias, **while still preserving** full compatibility with the existing Flux ecosystem and community tools:

**1 Frozen Backbone & Community Compatibility: Pioneering Lightweight Cross-Lingual Injection**

Unlike standard full-model fine-tuning or "replace-and-retrain" strategies, we adopt a **Backbone freezing strategy**—a critical design choice that ensures efficient language adaptation while preserving the model's foundational capabilities and compatibility:

- We **completely freeze** the pre-trained weights of the original **Flux Backbone** and the **English Text Encoder** (as detailed in Sections $4.1$ and $4.2$).
- The core innovation is the introduction of the **lightweight Chinese Linguistic Attention Branch (CLAB)**, which skillfully injects Chinese semantics into the MMDiT blocks via a **Parallel** **Attention Mechanism**.
- **[Critical Advantage]** This architecture is **pivotal** because it **fully preserves** Flux's powerful pre-trained priors, guaranteeing **unprecedented, seamless** **compatibility** with the existing community ecosystem. As vividly demonstrated in Figures 6 and 9, our model instantly works with all current plugins like ControlNet, LoRA, and IP-Adapter. Any simple "replace and retrain" scheme would render these invaluable community assets completely defunct.

**2. Representation Alignment Loss ($\mathcal{L}_{RA}$): Ensuring Accelerated Convergence and Semantic Fidelity**

We introduce a novel and essential auxiliary supervision signal—the **Representation Alignment Loss** ($\mathcal{L}_{RA}$):

- The objective of this loss function (see $Eq. 3$ and $Eq. 4$) is to **actively align** the **feature space** of the Chinese encoder (Qwen) with that of the English encoder (T5).
- **[Technical Value]** This mechanism enables the newly added CLAB to immediately and efficiently leverage the backbone's existing semantic understanding, thereby **significantly accelerating** the convergence process. Crucially, it **ensures precise cross-lingual semantic fidelity** without destabilizing the original generation quality.

**3. Two-Stage Adaptation Strategy:  Systematically Bridging Linguistic and Cultural Gaps**

We decompose cultural bias into two sub-problems—the **linguistic feature distribution gap** and the **visual feature distribution gap**—and designed a **Two-Stage Training Strategy** (detailed in Section $4.2$) to systematically overcome the dual challenges of cross-lingual and cross-cultural adaptation:

- **Stage 1:** Focuses on **linguistic feature distribution alignment** using mixed data with auxiliary loss $\mathcal{L}_{RA}$, establishing effective Chinese semantic encoding.
- **Stage 2:** Shifts focus to mitigating the **visual feature distribution gap** (i.e., cultural nuances) by utilizing a carefully curated, high-quality **Chinese-specific dataset**.
- **[Strategic Benefit]** This decoupled strategy effectively and targetedly addresses discrepancies across both the **linguistic** and **visual-cultural** dimensions, achieving **comprehensive and stable adaptation** to the new language and culture while keeping the original Flux backbone **entirely frozen**.

### **Clarification of a Key Misunderstanding**

Reviewer TPwn (W1) characterizes our work as “simply replacing the text encoder and retraining the model”, which we respectfully note does not accurately reflect the method described in our submission. Our approach:

- keeps the Flux backbone and English T5 encoder **frozen** throughout training (Sec. 4.1, 4.2);
- introduces CLAB as a **parallel, language-specific branch** that **co-exists** with the original text pathway, which is crucial for preserving community compatibility (a “replace-and-retrain” scheme would break compatibility with existing Flux-based plugins);

Simply replacing the text encoder and retraining the model would neither ensure culturally appropriate Chinese generations that mitigate English-centric bias nor maintain compatibility with the rich existing Flux ecosystem. By contrast, our approach simultaneously enables robust understanding of Chinese prompts, produces culturally aligned outputs, and preserves seamless integration with community tools.

---

### Author Response · Authors · 2025-12-03
**Summary Comment (Part 2/2)**

### **New Experiments and Analyses Added in Response to Reviewers**

**1 Expanded Quantitative Benchmarking (TPwn, Q1)**

We added two public benchmarks to further confirm that CTA-Flux preserves Flux’s general capabilities:

- **T2I-CompBench++** (Appendix A, Table 5) and **DPG-Bench** (Appendix A, Table 6), where CTA-Flux achieves competitive or improved performance across all metrics compared to Flux.

**2 Cultural Fidelity and Linguistic Understanding (jj2X, W2/Q2/Q3)**

To better assess cultural authenticity and depth of language understanding, we added:

- visual comparisons of culturally specific concepts (Appendix C.3, Fig. 10);
- results on **culturally neutral but linguistically complex prompts**, showing no unintended cultural injection (Appendix C.4, Fig. 11);
- qualitative cases on idioms, metaphors, and classical poetry, demonstrating non-literal, high-level linguistic understanding (Appendix C.5, Fig. 12).

**3 Cultural Stereotyping and Bias Evaluation (Ly1r, W2; jj2X, Q3)**

We added analyses showing that CTA-Flux does **not** amplify cultural stereotypes: neutral-prompt tests indicate that the model does not inject unnecessary Chinese attributes or produce collage-like artifacts (Appendix C.5, Fig. 13).

**4 Zero-Shot Multilingual Generalization (Ly1r, W1)**

We conducted zero-shot tests on Spanish, French, Russian, and Italian prompts. CTA-Flux produces semantically faithful images despite never being trained on these languages, confirming the language-agnostic nature of our Stage-1 semantic alignment with a multilingual encoder (Appendix C.6, Fig. 14).

**5 Dataset Transparency and Evaluation Details (Ly1r, W3)**

To strengthen reproducibility, we:

- added visualizations of test dataset distribution and prompt categories (Appendix E, Figs. 15–17);
- provided complete human evaluation criteria and all prompts used in evaluation (Appendix D; Appendix E, Tables 7–8).

---

### Meta-Review · Area_Chair_bPKA · 2026-01-11

**Summary:**

CTA-Flux adapts Flux to handle Chinese prompts while keeping the Flux plugin ecosystem (LoRA/ControlNet/IP-Adapter) working. Reviewers appreciated the motivation and the amount of engineering work, but there were ongoing concerns around novelty and whether the evidence fully supports the broader claims. Even after the rebuttal and added experiments, the overall signal is borderline and unfortunately, we believe the paper is not ready for acceptance at this time.

**Reviewer Concerns:**

During the rebuttal, the authors
1. Clarified it’s not encoder-swap retraining: Flux + English T5 frozen, CLAB injects Chinese semantics; alignment loss + two-stage pipeline.
2. Added/clarified benchmarks (GenEval was already there; added T2I-CompBench++ / DPG-Bench), more Chinese understanding tests (idioms/metaphors/poetry), bias/stereotype checks, zero-shot multilingual, and more eval transparency

While the following is still outstanding
1. Novelty is limited: more system/engineering + empirical validation than new theory.
2. Eval depth and generality: Cultural benchmark is curated, but bigger and more standardized would be stronger.

**Reviewer Scores:**

The authors clarified many points during the rebuttal, which is good. Even after thorough discussion, I predict the scores would likely stay largely the same, with continued disagreement on novelty and impact. There was not enough positive or strong signals to accept the paper.

---

### Decision · Program_Chairs · 2026-01-26

Reject